



# Insights from hailstorm track analysis in European climate change simulations

Killian P. Brennan[1], Iris Thurnherr[1], Michael Sprenger[1], and Heini Wernli[1]

[1]Institute for Atmospheric and Climate Science, ETH Zürich, Zurich, Switzerland,

**Correspondence:** Killian P. Brennan (killian.brennan@env.ethz.ch)

**Abstract.** Hailstorms are among the most destructive weather events, posing significant threats to infrastructure, agriculture, and human life. This study applies hailstorm-tracking diagnostics to kilometer-scale, decade-long climate simulations over Europe using the COSMO v6 model driven by ERA5 reanalyses. Convection is treated explicitly, and hail is modeled online with the HAILCAST parameterization. Simulations represent current and future climate simulations, the latter corresponding to a $3\,\mathrm{K}$ global temperature increase implemented via a pseudo-global warming approach.

We analyze high-frequency hail output at $5\,\mathrm{min}$ intervals, which enables tracking $\sim 40\,\mathrm{k}$ hailstorms in Europe in current and future climate simulations each. Storm track properties include length, duration, hail size, and spatial distribution, while three-dimensional environmental variables along these tracks yield storm-centered composites of hailstorm structure and allow for the examination of storm-inflow environments. Our analysis reveals significant shifts in the characteristics of hailstorms under the future climate scenario. Notably, hail frequency trends vary across Europe, but the trends in hailstorm environments are comparatively uniform. The most striking results are: (i) hail swath areas are projected to change both in terms of frequency and spatial extent, with a two-fold increased frequency of storms producing $\sim 50\,\mathrm{mm}$ and larger hail diameters. Per-storm hail swath areas generally expand by 15-30%, with swath area increases being more important for smaller hail, while frequency changes dominate for larger hail. (ii) The effect of increased hail melting due to the higher elevation of the $0\,^\circ\mathrm{C}$ level on the storm maximum hail diameters is found to be minor. (iii) Precipitation and wind hazards accompanying hailstorms are expected to increase on average by 20% and 5%, respectively, while extreme hail-precipitation compound events, i.e., hail with a diameter of at least $30\,\mathrm{mm}$ followed by $50\,\mathrm{mm}\,\mathrm{h}^{-1}$ are projected to be twice as frequent in the future.

## 1 Introduction

As one of the most destructive natural hazards, hailstorms lead to considerable economic losses and pose serious risks to public safety. In Europe, hail can cause extensive damage to crops, vehicles, buildings, and critical infrastructure, with annual losses estimated at several billion Euros (Punge and Kunz, 2016; Schmid et al., 2024). Hail can devastate entire harvests, impacting food supply and local economies (Portmann et al., 2024). Additionally, hailstorms can result in injuries or fatalities due to large hailstones and associated severe weather phenomena (Martín et al., 2024). Several atmospheric factors contribute to hail formation by promoting the development of strong convective storms. Weak static stability allows for vigorous upward motion in the atmosphere, which is essential for thunderstorm development (Markowski and Richardson, 2010). High low-level hu-



midity provides ample moisture for condensation, enhancing the energy of the storm through latent heat release and supporting stronger updrafts that can sustain hail growth (Wallace and Hobbs, 2006). Additionally, vertical wind shear organizes storms into longer-lived convective cells like supercells, which are more likely to produce hail due to their intense and sustained updrafts (Doswell, 2001).

In the context of climate change, three competing processes governing hail growth are most relevant to consider. First, a warmer atmosphere can hold more moisture in accordance with Clausius-Clapeyron scaling (Trenberth et al., 2003; Held and Soden, 2006; Seneviratne et al., 2021), which can lead to more intense convection with stronger updrafts and ultimately larger hailstones. Second, assuming unaltered lapse rates, the $0\,°C$ level is higher in a warmer climate, which would give the hailstones more time to melt as they fall and result in smaller hailstones at the surface. This effect is expected to be larger for

smaller hailstones as they have a lower terminal velocity compared to large hailstones. And third, regional climate simulations over Europe indicate that the future summer warming pattern under climate change is affected by spatially differing lapse-rate changes, where steeper lapse rates would reduce stability, promoting deep convection and shallower lapse rates increasing stability and suppressing convection (Brogli et al., 2021).

Raupach et al. (2021) provided a comprehensive summary of the impact of climate change on hailstorms globally, predicting

an increased frequency of hailstorms in Australia and Europe while expecting a decrease in East Asia and North America. The study also anticipated an increase in hailstorm severity worldwide (i.e., larger hailstones), due to rising low-level moisture and convective available potential energy. Elevated $0\,°C$ level height and the associated increase in hail melt are expected to enlarge the average hailstone size, while the effect of decreased vertical wind shear on hailstorm activity was estimated to be limited. However, this review also highlighted significant uncertainties due to limited long-term data and the need for high-resolution,

convection-permitting numerical modeling to better understand current hailstorm climatologies and future hailstorm changes.

Climatological trend information about hail and/or the environment of hailstorms can be obtained through various means: (i) Trend analysis with radar-based hail climatologies, which provides observational data but lacks information about the hail environment (e.g., humidity and temperature profiles). (ii) Hail proxies derived from reanalysis data, considering only hail precursors but not actual hail formation. Early studies used environmental proxies to estimate hail occurrence also from

coarse-resolution climate simulations (e.g., Kapsch et al., 2012; Sanderson et al., 2015; Rädler et al., 2019). (iii) Online hail parameterization in climate simulations, which is the approach we use in this study. Diagnostic, physics-based models such as HAILCAST (Brimelow et al., 2002; Adams-Selin and Ziegler, 2016) have been employed either offline (Brimelow et al., 2017) or online (Cui et al., 2024) to quantify hail diameters expected at the ground. Similarly, hail occurrence has also been inferred from vertically integrated graupel (e.g., Trapp et al., 2019). (iv) Convection-permitting simulations of hail with explicit hail

microphysics (e.g., Gensini et al., 2024), which are not currently feasible to run over the entire European domain for decadal time periods due to computational resource limitations.

Several methods to quantify hail occurrence in climate model simulations have been explored in the past. Studies that investigated future trends in hail occurrence in the U.S. and Europe are summarized in Tab. 1. Object-based analysis involves identifying and tracking individual weather phenomena as coherent, typically two-dimensional, entities, offering quantitative

information about their life cycle and structural evolution, which is not available from standard Eulerian analysis. This approach



**Table 1.** Overview of previous numerical modeling studies on climate change effects on hail that feature periods of more than $10\,\mathrm{y}$. The entries provide, from left to right, information about the simulated region, the horizontal grid spacing, the number of horizontal grid-points, the type of method used to infer about hail, the simulated period, and a flag about hailstorm tracking.

| | region | $\Delta x$ | grid-points | hail | period | tracking |
|---|---|---|---|---|---|---|
| Kapsch et al. (2012) | Germany | 50 km | — | proxy | $4 \times 15\,\mathrm{y}$ | no |
| Sanderson et al. (2015) | British Iles | 25 km | — | proxy | $128\,\mathrm{y}$ | no |
| Brimelow et al. (2017) | U.S. | 50 km | — | offline diagnostic | $2 \times 30\,\mathrm{y}$ | no |
| Prein et al. (2017a) | U.S. | 4 km | $1360 \times 1016$ | — | $2 \times 13\,\mathrm{y}$ | yes |
| Trapp et al. (2019) | U.S. | 4 km | — | graupel | $2 \times 30\,\mathrm{y}$ | no |
| Rädler et al. (2019) | Europe | 50 km | — | proxy | $3 \times 30\,\mathrm{y}$ | no |
| Ashley et al. (2023) | U.S. | 3.75 km | — | — | $2 \times 15\,\mathrm{y}$ | yes |
| Gensini et al. (2024) | U.S. | 3.75 km | — | explicit | $5 \times 15\,\mathrm{y}$ | no |
| Thurnherr et al. (2025) | Europe | 2.2 km | $1542 \times 1542$ | online diagnostic | $2 \times 11\,\mathrm{y}$ | no |

is well-established in cyclone tracking studies (Hodges, 1999; Neu et al., 2013) and has been applied to convective storms without a specific focus on hail (Davis et al., 2006). As demonstrated by Prein et al. (2017b), Ashley et al. (2023), and Brennan et al. (2024), applying object-based analysis to convective storms provides valuable insights into the characteristics of mesoscale convective systems, enhancing our understanding of their dynamics and environmental conditions. So far, not many

climate simulation studies focusing on convection have made use of object-based analysis, and so far none have investigated hailstorms in climate simulations through this approach (Table 1). One reason for this might be that this type of analysis requires very high-frequency output from climate simulations. In this study, we make use of high-frequency hail diameter outputs (every 5 minutes) from high-resolution, convection-permitting simulations (Cui et al., 2024; Thurnherr et al., 2025), which allows for a detailed tracking of individual hailstorms and an object-based analysis of current and future climate hailstorms in Europe.

The simulations apply a pseudo-global warming approach (PGW, Brogli et al., 2023) to simulate the effect of a 3 K global warming level on convective systems over Europe including the online hail diagnostic HAILCAST to simulate hail diameter. According to these simulations, the 3 K global warming level leads to an increase in the number of hail days over northeastern Europe and a decrease over southwestern Europe during summer due to an increase, respectively a decrease, in convective available potential energy and low-level moisture content (Thurnherr et al., 2025).

Building on these results and using the same simulations, our study aims to assess the impact of global warming on hailstorm dynamics by integrating advanced numerical climate modeling with object-based analysis. We use a recently introduced hailstorm tracking (Brennan et al., 2024) to study the effect of anthropogenic climate change on hailstorm environmental conditions. We quantify how environmental conditions surrounding hailstorms differ under future climate conditions. Specifically, we aim to answer the following questions:

1. How does the lifetime and length of hailstorm tracks change with 3 K global warming?



2. How does the propagation speed of hailstorms change?

3. Does the hail swath area change and how does the answer to this question depend on hail size?

4. Are there changes in the magnitude and structure of convective available potential energy (CAPE), convective inhibition (CIN), vertical wind shear, specific humidity, and potential temperature in the environment of the hailstorms?

5. Do the effects of climate change on hailstorm characteristics and environments geographically vary across Europe?

6. How do the storm-associated hazards precipitation and wind change along the hailstorm tracks?

To this end, the spatial distribution of storms and their characteristics are investigated (Sect. 3), using an object-based approach (Sect. 2.3). Additionally, environmental conditions surrounding hailstorms are analyzed through storm-centered composites (Sect. 4 and 5). Finally, the findings are summarized and conclusions are drawn in Sect. 6.

## 2 Methods and data

In this section, an overview of the methods and data utilized in this study is provided. The numerical climate simulations using COSMO v6 are detailed in Sect. 2.1, along with specifications on the hail parameterization HAILCAST in Sect. 2.2. This is followed by a description of the storm tracking algorithm in Sect. 2.3.

### 2.1 COSMO climate simulations

This study builds on the same COSMO model simulations as discussed in Thurnherr et al. (2025). For the current climate, simulated precipitation and hail were rigorously validated against different types of observations, as documented in Cui et al. (2024). COSMO v6 is a limited-area numerical climate model, driven in this study at the lateral boundaries by ERA5 reanalysis data (Hersbach et al., 2020) for the current climate scenario. To project future climate conditions, a PGW approach was employed, incorporating a 3 K increase in global mean temperature (Schär et al., 1996; Hara et al., 2008; Leutwyler et al., 2016; Brogli et al., 2023). The specifics of the PGW simulation were described in Thurnherr et al. (2025). Both simulations were run for 11 years, covering the period 2011-2021. This method allows for a systematic comparison of hailstorm characteristics under current and projected future climate simulations, providing insights into the potential impact of global warming on hailstorm characteristics and environments. The control and PGW simulations are referred to as *current climate* and *future climate* simulations hereafter.

The COSMO model, a non-hydrostatic, limited-area numerical weather prediction (NWP) model, solves the equations governing compressible flow in a moist atmosphere on a rotated-pole grid with hybrid terrain-following height coordinates (Steppeler et al., 2003). Originally developed for operational NWP, the COSMO model has been widely used in scientific research at meso-$\beta$ and meso-$\gamma$ scales, particularly for convection-permitting weather forecasts (Baldauf et al., 2011; Klasa et al., 2018) and regional climate simulations (e.g., Ban, 2015). The simulations used in this study operate at a high resolution with a horizontal grid spacing of $2.2\,\mathrm{km}$, enabling a fairly detailed representation of severe convective weather phenomena over the Alpine



region on climatological time scales (Prein et al., 2013; Ban et al., 2014). This convection-permitting resolution is crucial for accurately capturing small-scale processes, such as organized, rotating thunderstorms, which are essential for understanding hailstorm dynamics. The two-dimensional fields of precipitation and hail (Sect. 2.2) were saved every $5\,\mathrm{min}$, while other two- and three-dimensional fields (at 8 pressure levels) were output hourly.

## 2.2  HAILCAST

CAM-HAILCAST (referred to as HAILCAST in this study) is a diagnostic hail growth parameterization that predicts maximum hailstone diameters reaching the ground and is designed for embedding in convection-permitting atmospheric models (Adams-Selin and Ziegler, 2016). It employs a one-dimensional cloud model coupled with a hail growth model, simulating processes such as liquid water accretion, ice particle collection, and sublimation while differentiating between wet and dry growth regimes. Notably, the horizontal advection of hailstones is neglected. This study focuses on the hail size as estimated from inserting a $10\,\mathrm{mm}$ embryo. For further details on how HAILCAST is implemented in the model, we refer to Cui et al. (2024).

## 2.3  Hailstorm tracking algorithm

Hailstorms are comparatively small-scale weather systems that can propagate at high speeds. This makes it challenging for tracking tools to maintain continuity, as these features may not spatially overlap between consecutive model output time-steps. This study's storm tracking algorithm, introduced by Brennan et al. (2024), first detects hailstorms as two-dimensional objects, defined by grid-points where the hail diameter surpasses a threshold of $10\,\mathrm{mm}$, and then tracks these objects in time, using various thresholds and parameters.

The algorithm begins by identifying features as local maxima in simulated hail diameter. These features are then grown in two dimensions until the threshold diameter is reached and subsequently segmented using a watershed algorithm. Objects with $d_{\mathrm{hail}} > 10\,\mathrm{mm}$ were tracked, and those with a prominence greater than $10\,\mathrm{mm}$ or maxima separated by more than 6 grid-points, were divided using a watershed algorithm. The area threshold was set to 9 grid-points, and finally the storm mask was expanded by 1 grid-point through binary dilation to enhance tracking robustness. The algorithm anticipates forward movement by shifting labeled areas based on a weighted mean of previous movement vectors. Correspondence between features across time-steps is determined by calculating tracking probability scores, with mechanisms to handle complex feature interactions such as splitting and merging.

Further, the tracks help mitigate the 'fishbone effect', which arises from the discontinuous hail swaths caused by low temporal sampling of fast-moving storms. To this end, a gap-filling method interpolates storm footprints between time steps, translating them along the storm's movement vector to reconstruct a continuous swath while preserving small-scale details. More details about the gap filling and the tracking algorithm can be found in Brennan et al. (2024).

Appendix A explains how the bootstrapping methodology was employed to grid the hail tracks.



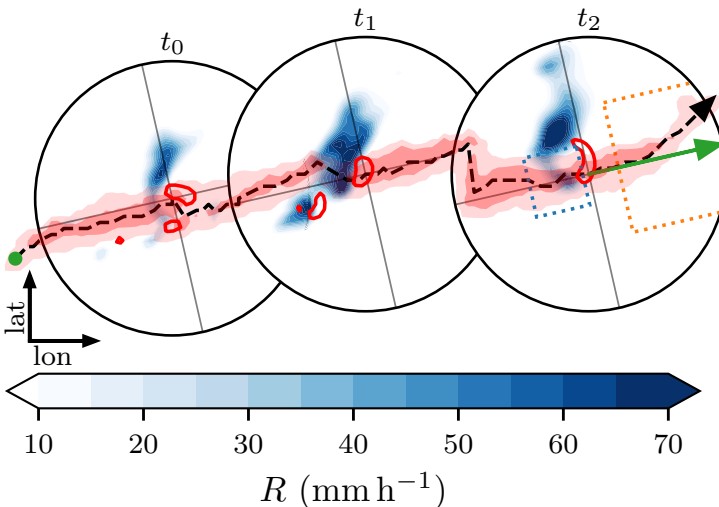

**Figure 1.** Schematic view of three footprints (black circles) extracted along an example storm track at intervals of 1 h. Blue shading shows the 5 min rain rate, red shading shows the extent of the $d_{\text{hail}} > 10\,\text{mm}$ and $20\,\text{mm}$ hail swath, red contours indicate $10\,\text{m s}^{-1}$ updraft at $400\,\text{hPa}$, and the black dashed line and arrow reveal the storm track and direction of movement, respectively. The green arrow indicates the mean propagation vector (over the entire storm lifetime) and the crosshairs reveal the orientation of the footprints relative to their propagation and the footprint centers. This crosshair is also featured in the composite plots. The start of the storm track is indicated with a green circular marker. The orange dotted line indicates the extent of the inflow sector of the storm, which is a $50{\times}50\,\text{km}^2$ box, $10\,\text{km}$ ahead of the storm center, while the blue dotted box, which is a $20{\times}20\,\text{km}^2$ box, $2\,\text{km}$ behind the storm center is used for rain-after-hail assessment.

### 2.4 Storm-centered composites

For all tracked cells (Sect. 3) with lifetimes $\geq 1\,\text{h}$, atmospheric fields within a radius of $r = 60\,\text{km}$ of the storm center were extracted at every full hour of storm evolution (Fig. 1). These extracted footprints were then rotated so that their respective storm-mean propagation vectors aligned with the abscissa. The atmospheric fields of the extracted footprints are later used to construct storm-centered composites (Sect. 4) and assess the response of the hailstorms to the inflow environment (Sect. 5).



## 3 Hailstorm tracks

In this section, we discuss the results of tracking hailstorms in the climate simulations using the algorithm introduced in Sect. 2.3. We analyze the spatial distribution of hailstorm tracks, as well as the characteristics of these storms, including

maximum hail diameter, mean storm area, lifetime, and propagation velocity. In the 11 simulated years, 39 594 and 37 746 (−4.7%) hailstorms with lifetimes ≥1 h were tracked within the model domain for current and future scenarios, respectively (see Tab. S1), of which 671 and 711 occur in DJF, 4 425 and 4 439 in MAM, 25 289 and 22 393 in JJA, and 9 209 and 10 203 in SOM for current and future climate simulations respectively. In the next paragraphs, we explore the characteristics of these tracked storms.

Generally, in line with the Eulerian analysis in Thurnherr et al. (2025), the storm tracks in both the current and future simulations in summer (June, July, August, JJA) are mostly constrained to the land, with only a few tracks over the ocean (Fig. 2a,b). In the current climate simulation, the maximum density of tracks is located just south of the main Alpine crest in the Po Valley (Fig. 2a). A distinct minimum of less than 5 storms per summer follows the main Alpine crest from southeastern France to Austria. The maximum in the future simulation is shifted to the east, towards northeastern Italy, southern Austria, and

Slovenia (Fig. 2b). This shift is also evident in the difference field, with a positive significant change in hail track density along the eastern Alps with the bulk in Austria (Fig. 2c). The hail track density is significantly reduced in the band spreading from the Iberian Peninsula, through France, northwestern Italy, northern Germany, and northern Poland. In spring (MAM), only very few hailstorm tracks were found in both the current and future climate simulations. These tracks mainly occur in eastern Europe (Fig. S1a,b), but due to the low numbers, trends are difficult to investigate (Fig. S1c). In autumn (SON), the hailstorm

tracks are predominantly located over the Mediterranean Sea (Fig. S1g,h), and trends in the number of storm tracks are mostly positive (Fig. S1i). Our analysis reveals that the spatial distribution of hail tracks in the COSMO v6 current-climate simulation aligns well with the observed climatology in the Alpine region, as reported by Nisi et al. (2018) and discussed extensively in Cui et al. (2024). The spatial distribution of the hailstorm tracks with lifetimes of 1 h and more are similar to that of supercells in the same simulations (Feldmann, 2025), with some exceptions. In current climate simulations, the supercell distribution

features a less pronounced hot spot in the Po Valley accompanied by a less distinct inner-Alpine minimum compared to the hail storm density (Fig. 2a). In future climate simulations, there are again more supercells in the inner-Alpine region, whereas the hail cells retain a minimum along the main Alpine crest (Fig. 2b).

Mean storm maximum hail diameters are typically in the range of 18 to 25 mm throughout Europe. The geographical distribution of mean storm maximum hail diameters lacks any prominent features, with the exception of some local maxima

along the North Sea coast in the future simulation (Fig. 2d,e). However, the mean of storm maximum diameters is projected to increase significantly in most of Europe by +1 to +4 mm, with only a few areas showing weak negative trends (along the Pyrenees, the English Channel, and the North Sea coast; Fig. 2f). Across the domain, the climate signal for the mean storm maximum hail diameter is +3.6% (Fig. S3c).

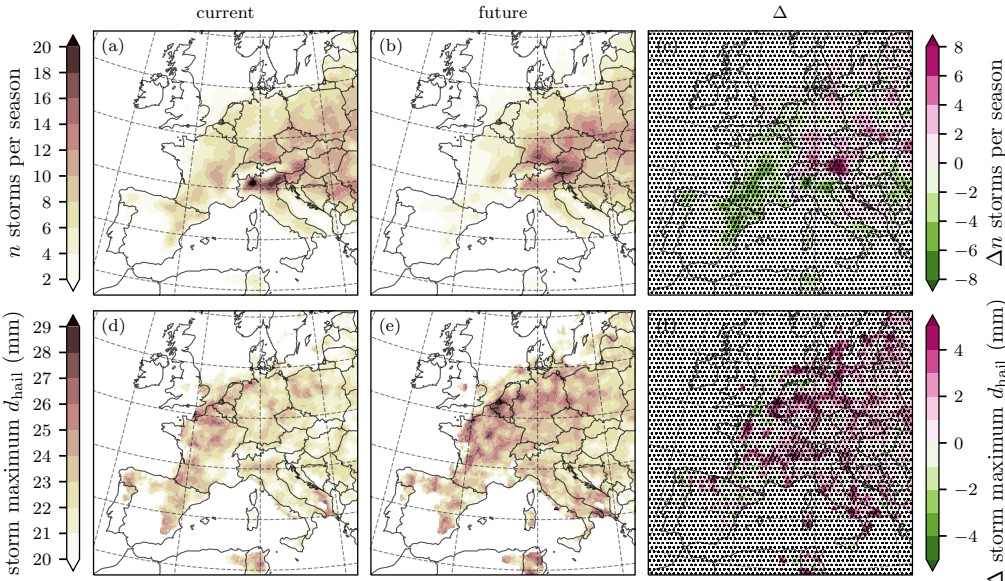

**Figure 2.** Number of tracked storms per season with lifetimes $\geq 1\,\mathrm{h}$ in JJA in the (**a**) current and (**b**) future climate ($n = 25\,289$ and $22\,393$ respectively). Values correspond to the number of storms within a circle with radius $r = 50\,\mathrm{km}$. (**c**) shows the difference of (b)-(a), i.e., between future and current climate simulations (filled contours); dots indicate regions where the difference is significant ($> 2\sigma$ according to $n = 1000$ bootstrap samples, see Sect. A). Panels in the second row (**d–f**) are organized identically to the ones in row one but for maximum hail diameter (averaged over storms within circle of radius $r$). In all panels, values are shown only at grid points with more than 3 storms per season. The latitude and longitude grid lines have spacings of $5°$ and $10°$, respectively.

Furthermore, mean instantaneous storm areas have values ranging from $100$ to $280\,\mathrm{km}^2$ and they exhibit no distinctive features in the overall spatial distribution, with the exception of a local maximum in central Germany in the future climate (Fig. 3a,b). By contrast, significant positive differences of $+40\,\mathrm{km}^2$ prevail across most of Europe (Fig. 3c). Averaged across the domain, the climate signal for the mean storm area amounts to $+10.3\%$ (Fig. S3b).

The distribution of mean storm lifetimes exhibits only weak spatial features across the domain (Fig. 3d,e). Additionally, changes in storm lifetime have opposite signs in close proximity, indicating high variability and most likely no robust signals in the relatively short simulations. Only some larger areas in eastern France, Germany, and in parts of eastern Europe show coherent positive, significant signals (Fig. 3f). Within the whole domain, the climate signal for the mean storm lifetime is $+1.7\%$ (Fig. S3a), and therefore comparatively weak.



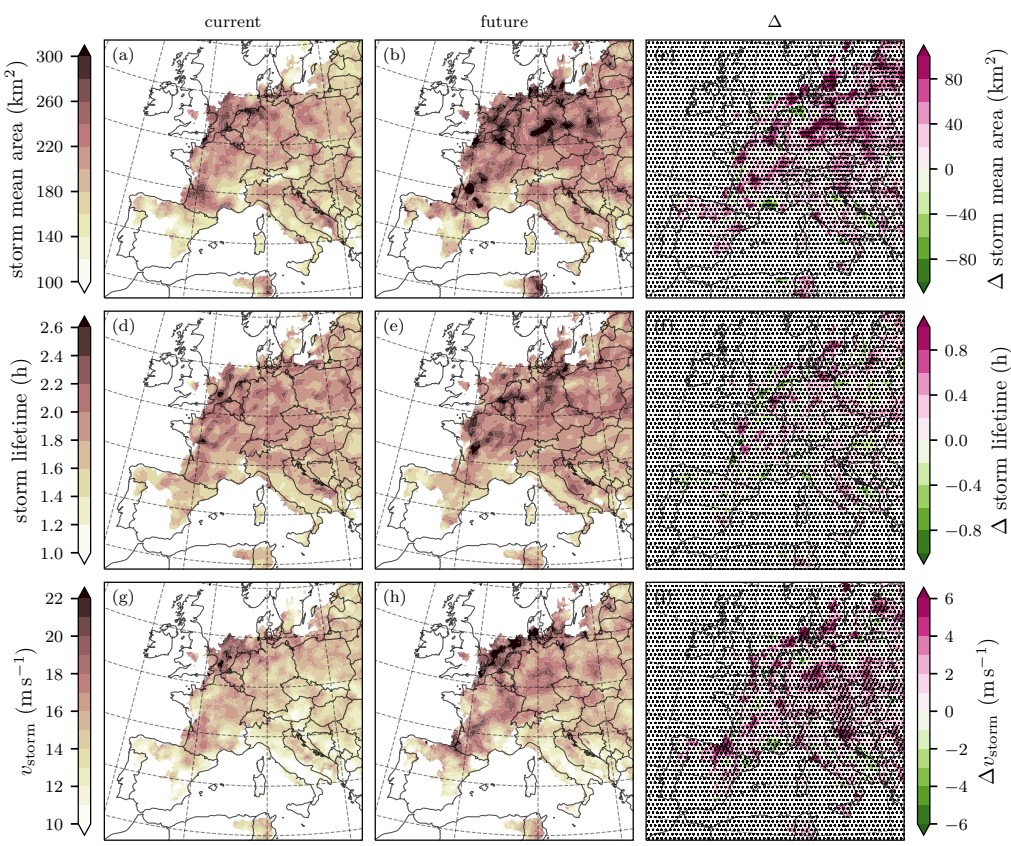

**Figure 3.** As Fig. 2, but for (**a–c**) mean storm area, (**d–f**) mean storm lifetime, and (**g–i**) mean storm propagation velocity.

Further, mean storm propagation velocity exhibits a weak south-to-north gradient, with higher velocities further north (Fig. 3g,h). Changes in storm propagation velocity are overwhelmingly positive with values in the range of $+1$ to $+3\,\mathrm{m\,s^{-1}}$.
These positive changes are significant in regions throughout the domain (Fig. 3i). Spatially averaged, the climate signal for the mean storm propagation velocity is $+7.3\%$ (Fig. S3d). Since storm lifetimes remain largely unchanged, the increase in propagation velocity implies that storms cover larger areas throughout their existence.



These gridded visualizations of storm track density illustrated the spatial variability in storm activity and the effects of climate change, with notable increases in track density in northeastern Italy, southern Austria, and Slovenia under future conditions. These results are largely consistent with the tracking-independent Eulerian hailstorm frequency analysis in Thurnherr et al. (2025). By employing tracking methods, the analysis of storm characteristics, including maximum hail diameter, mean storm area, lifetime, and propagation velocity, provides insights that go beyond traditional Eulerian analysis and allow for a more nuanced understanding of storm evolution and variability under future climatic conditions. The observed trends, such as the increase in mean maximum hail diameter and storm area, and the predominantly positive changes in storm propagation velocity, underscore the importance of understanding the behavior of individual storms under changing climatic conditions. The changes in these properties are spatially relatively uniform. The tracking-based analysis highlights that, while the change in hail track frequency varies regionally, the hailstorm property changes are surprisingly consistent across regions. Such findings are less apparent in Eulerian approaches, emphasizing the added value of storm tracking for dissecting storm behavior. These results hint towards a different effect of climate change on storm initiation but similar effects on storm evolution.

A critical aspect of hailstorm impacts is the spatial extent of hailfall, which determines the overall affected area. In the following, we investigate how climate change influences the per-storm hail swath area and relate this to hail storm frequency changes. To this end, we investigate how the hail swaths (i.e., per storm time-integrated area affected by a certain hail-diameter threshold, see Fig. 1) change between hailstorms in current and future climate simulations. Specifically, we compare the gap-filled (see Sect. 2.3) hail swath area as a function of hail diameter and the corresponding frequency distributions between the two simulations. This analysis reveals a slight decrease in the frequency of storms with a maximum hail diameter of $\leq 30\,\mathrm{mm}$ and a more increased frequency above that, while a two-fold increase of hailstorms with maximum diameters above $\sim 50\,\mathrm{mm}$ is simulated (Fig. 4a).

Per-storm hail swath areas increase for all but the smallest hail diameters ($\leq 11\,\mathrm{mm}$), while the largest positive change in hail swath area is simulated for the $15\,\mathrm{mm}$ diameters ($+38\%$, Fig. 4b). Swath areas for hail diameters between $20\,\mathrm{mm}$ and $40\,\mathrm{mm}$ increase by $16 - 21\%$. For hail diameters beyond $50\,\mathrm{mm}$, the increase in swath area is again substantial, but also more uncertain due to the smaller sample size ($> 30\%$, Fig. 4b). When considering the combined impact resulting from changes in frequency and swath area, the changes due to increasing swath area are more important for hail diameters below $40\,\mathrm{mm}$, while for larger diameters, changes in storm frequency dominate (Fig. 4a,b).

This analysis was repeated for adapted PRUDENCE regions, as documented in the Supplement (Christensen and Christensen, 2007, see Fig. S2,S17–S28). Most regions follow the behavior of the entire simulation domain with respect to hail swath frequencies and areas (Fig. 4a,b), with some distinct exceptions. Notably, the Baltic region features a projected increase in frequency and area of more than 30% across all but the smallest hail diameters (Fig. S18b). Further, swath frequencies and areas in the British Isles remain mostly unchanged under climate change (Fig. S19a,b). France and the Iberian Peninsula exhibit reduced frequencies for most hail diameters accompanied by a lack of increased area for large hail diameters (Fig. S22a,b). Projections for the Mediterranean region reflect a slightly reduced hail swath area for larger hail diameters (Fig. S25a,b).



The hail swath area analysis reveals changes in the combination of area and hail diameter that are not directly evident from Eulerian methods. The multifold increase in hail-affected areas of hailstorms in the future climate simulations — particularly evident for the large hail diameters — is most important to keep in mind when assessing future hail risk. This increase suggests that hail-producing storms in the future will not only be more intense but will also impact a significantly larger area, which
could exacerbate damage to infrastructure, agriculture, and insurance costs. Understanding these spatial changes is critical for improving risk assessment models and developing adaptation strategies to mitigate the effects of more widespread hailstorms. To this end, we present an extensive dataset of hail swaths compiled within this study, which is well-suited for stochastic resampling techniques (e.g., Schröer et al., 2023) and can be effectively utilized as a hazard component in hail damage modeling (e.g., Schmid et al., 2024). By leveraging this dataset, researchers and policymakers can gain deeper insights into the evolving
risk landscape of hailstorms and refine mitigation strategies. It includes 39 594 and 37 746 hail swaths for current and future climates, respectively. The dataset is available as explained in the *Code and data availability* section.

In order to examine the potential factors that influence shifts in hailstorm characteristics under a warming climate, we focus on storm-centered composites of the most intense events (Sect.,4).

## 4  Storm-centered composites

We now investigate storm-centered composites, which focus on the central point in space of each storm footprint and aggregate various meteorological variables around these points, providing a detailed spatial and temporal view of the storms' immediate vicinity, both for the current and future climate scenario. By centering the analysis on the storms themselves, we can accurately capture the atmospheric conditions associated with hailstorm genesis, evolution, and dissipation, as well as the spatiotemporal structure of the storms. To characterize the environment of the storms, we consider in particular temperature, specific humidity,
winds, and CAPE[1]. The storm-centered composites thus provide a complementary perspective to the track-based analysis, enabling precise identification of the environmental signatures that accompany hail-producing storms.

A total of $n = 31\,899$ and $28\,259$ footprints were extracted for current and future scenarios in JJA, respectively (see Sect. 2.4 for details). These footprints were further filtered, to only consider footprints of the most intense storms during their most intense period[2]. To this end, the 10% of all-season storm-centered footprints yielding the largest hail diameters were chosen for
constructing the composite ($n = 3641$ and $3284$ footprints for current and future climate simulations, respectively, in the entire simulation domain in JJA; see Table S1 and Fig. S4). Our approach is similar to previous studies that presented composites of convective storms (e.g., Prein et al., 2017a; Brennan et al., 2024; Arnould et al., 2025).

---

[1]As inflow maximum mixed layer CAPE and most unstable CAPE correlate well ($c = 0.94$), we would expect the analysis involving inflow CAPE to hold true for both choices of CAPE (on average, most unstable CAPE is $1.25\times$ larger than mixed layer CAPE). We use exclusively mixed layer CAPE and CIN in this study.

[2]A note on filtering the composite constituents: Filtering footprints every hour by the $n$ largest hail diameters is not equivalent to filtering by the $n$ storms that yield the largest hail diameters. The second approach is arguably the better approach as the hourly sampling might miss the most intense part of the storms yielding the largest diameters. The representation of the $n$ footprints with largest hail diameters essentially gives the storm structure during the most intense phases of the storms as seen by the hourly sampling.

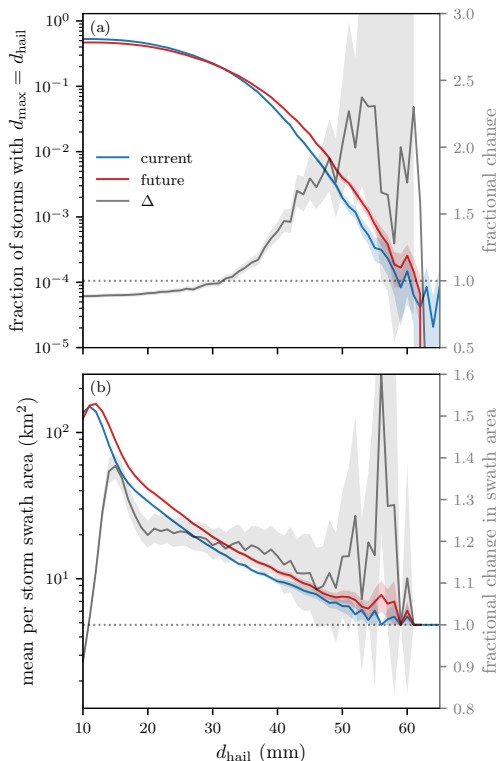

**Figure 4.** Hail swath statistics as a function of maximum hail diameter for all hail events simulated in JJA: (**a**) fraction of storms that exhibit a particular maximum hail diameter in current (blue) and future (red) climate simulations ($n = 25\,289$ and $22\,393$ storms respectively) and the fractional change between the two periods (grey). (**b**) Mean per-storm swath area for a given maximum hail diameter in current (blue) and future (red) periods. The grey lines show relative changes in the swath area impacted by respective hail diameters. Computations for both panels are performed for 1 mm wide bins and shaded areas show the 5–95[th] percentile range of $n = 1000$ bootstrapped samples of the gap filled hail swaths (see Sect. 2.3). Equivalent figures for the different subdomains are included in the Supplement (Fig. S17 – S28).

### 4.1 Current climate

From the storm-centered composites of the current-day simulations, the archetypal hailstorm structure emerges (Fig. 5). At
ground level, the potential temperature field exhibits a pronounced gradient of $4\,\mathrm{K}$ over $25\,\mathrm{km}$, oriented $45°$ from the direction of storm movement, with a minimum situated just behind the hail shaft (Fig. 5f). This temperature minimum aligns with a




pressure maximum, both resulting from downdraft air drawn in by intense precipitation and evaporative cooling. The wind vector field reveals near-ground divergence at this point, where the downdraft transitions into horizontal wind at the surface (Fig. 5c). Approximately $20\,\mathrm{km}$ ahead of the storm, a convergence area is noticeable in the $10\,\mathrm{m}$ wind field, indicating the onset of the updraft column.

At the inflow level, around $850\,\mathrm{hPa}$, there is a maximum of specific humidity where air converges with a cyclonic component in the absolute horizontal wind field, reaching a value of $12.8\,\mathrm{g\,kg^{-1}}$ (Fig. 5e). Higher up, at $400\,\mathrm{hPa}$ where the updraft core is situated (Fig. 5e), the horizontal winds near the storm are influenced by the synoptic situation, primarily determined by the horizontal pressure gradient and largely unaffected by the storm. On average, the storm's updraft core at this altitude spans no more than $10\,\mathrm{km}$. Since storm tracking is conducted on the hail field, which is highly congruous to the vertical wind maxima, the vertical wind signatures align well, resulting in a well-defined composite structure. The co-location of hail and updraft maxima is expected, as HAILCAST does not consider the horizontal advection of hailstones.

Approximately $5\,\mathrm{km}$ behind the storm center, a peak in rainfall occurs, with mean rates reaching $35\,\mathrm{mm\,h^{-1}}$ (Fig. 5b). The precipitation footprint is asymmetric, extending further to the left relative to the storm's motion. Conversely, the outline of cloud water slightly leads the storm center due to the vertical shear (Fig. 5b). Graupel is explicitly included in the COSMO microphysics and is subject to horizontal advection, which, however, results in only a slight offset of the graupel maximum from the storm center to the left relative to cell movement (Fig. 5a). The location of the graupel maximum provides an upper limit on potential hail advection, as graupel has a lower terminal velocity than the smallest hailstones, allowing more time for horizontal advection.

Finally, we consider CAPE values, which decrease rapidly as the storm approaches. $25\,\mathrm{km}$ ahead of the storm, CAPE values reach $1600\,\mathrm{J\,kg^{-1}}$ on average, but they drop below $400\,\mathrm{J\,kg^{-1}}$ just as the storm passes (Fig. 5d). The CAPE field exhibits an asymmetry and the largest values are located to the right (relative to the movement direction of the storm), in front of the updraft. The thunderstorm anvil cloud extends beyond the $40\,\mathrm{km}$ window size ahead of the storm (Fig. 5d).

## 4.2 Climate change signal

To investigate the mechanisms driving the intensification of hailstorms in the future simulations during JJA, we examine differences in hailstorm environments and dynamics between the two climate simulations. Based on the results presented in Sect. 3, a consistent increase in composite mean hail diameter is expected, with the largest changes ($+1.2\,\mathrm{mm}$) close to the storm center (Fig. 6a). The increases in rain rates ($+19.5\%$, Fig. 6b) and $10\,\mathrm{m}$ wind gusts ($+5.2\%$, Fig. 6c) around the storm center indicate a substantial change, with future storms potentially characterized by more intense rainfall and stronger winds, enhancing both the hydrological impact and wind damage potential of these events.

The increase in CAPE across the storm's environment (Fig. 6d) is particularly noteworthy, as it implies a significant boost in the energy available for storm intensification, likely leading to more severe convective phenomena. This observation is consistent with the Eulerian analyses of the same simulations by Thurnherr et al. (2025). Interestingly, they noted a decrease in mean seasonal CAPE in regions of southwestern Europe experiencing a decline in hail frequency. However, even with a mean decrease in CAPE, exceptionally high CAPE values can still occur in the specific situations when the environment favors

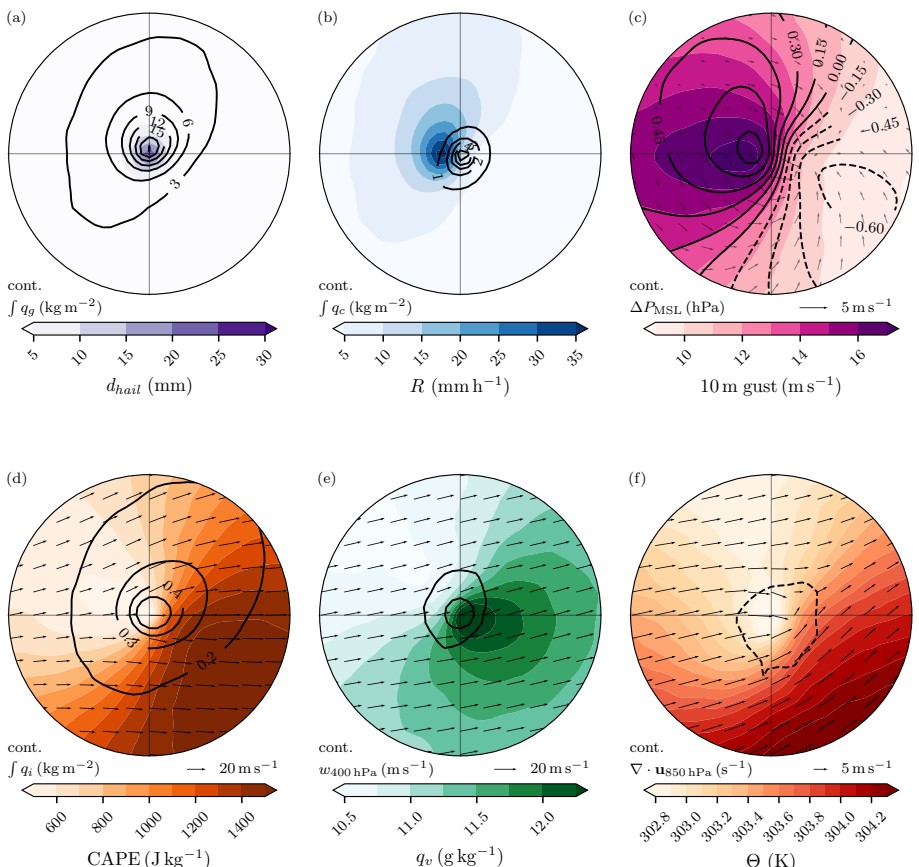

**Figure 5.** Composite analysis of most intense storm footprints (90[th] percentile of hail diameter, $n = 3641$) in the current climate in JJA, centered on their footprint center and rotated so that their respective movement vector aligns with the $x$-axis (with the storm moving to the right). The radius of the panel outline is $40\,\mathrm{km}$. Shown are (**a**) HAILCAST maximum hail diameter (filled contours) and column integrated graupel (black contours); (**b**) hourly rain rate (filled contours) and column-integrated cloud water (black contours); (**c**) $10\,\mathrm{m}$ wind gusts (filled contours), mean sea level pressure anomaly (black contours), and $10\,\mathrm{m}$ absolute wind field (quivers); (**d**) CAPE (filled contours), column-integrated cloud ice (black contours), and $200\,\mathrm{hPa}$ absolute wind field (quivers); (**e**) specific humidity at $850\,\mathrm{hPa}$ (filled contours), vertical wind at $400\,\mathrm{hPa}$ (1 and $5\,\mathrm{m\,s^{-1}}$, black contours), and horizontal absolute wind field (quivers) at $400\,\mathrm{hPa}$; and (**f**) potential temperature at $850\,\mathrm{hPa}$ (filled contours), horizontal divergence at $850\,\mathrm{hPa}$ ($-0.1\,\mathrm{s^{-1}}$, black contour), and horizontal absolute wind field (quivers) at $850\,\mathrm{hPa}$. The same plot for the most intense storm footprints in the future climate can be found in the Supplement (Fig. S5).



hail formation. This is corroborated by the uniform increase in specific humidity at 850 hPa in the storm-centered composites (Fig. 6e), which supports enhanced moisture availability — a key factor in storm development and intensity. The consistent rise in potential temperature at 850 hPa (Fig. 6f) across the entire storm-influenced area further suggests a warmer and potentially less stable lower atmosphere conducive to storm formation and persistence. Future climate simulations produce hailstorms with inflow areas (see Fig. 1) that are $1.83\,\mathrm{K}$ warmer (at $850\,\mathrm{hPa}$) than in current climate simulations. These changes, distinguished by their statistical significance, underscore the anticipated shift towards more hazardous storm conditions in the future climate simulation.

Liquid precipitation following hail can be crucial for assessing storm damage, as rain can penetrate buildings through hail-damaged skylights, windows, and roofs, causing additional damage via water ingress (Ridder et al., 2020). Our analysis indicates that liquid precipitation typically follows hail immediately after the passage of the storm center, with high intensities exceeding $20\,\mathrm{mm\,h^{-1}}$ (Fig. 5a,b). In fact, the fraction of storms with potentially damaging $>20\,\mathrm{mm}$ hail that feature a mean precipitation rate of $>20\,\mathrm{mm\,h^{-1}}$ in a $20{\times}20\,\mathrm{km^2}$ box behind the hail shaft (rain-after-hail, see Fig. 1) is projected to increase from 13.4% to 18.5% by (relative change of +36.1%). This increase is even greater for higher precipitation rates (i.e., almost +80% for intensities $>50\,\mathrm{mm\,h^{-1}}$, Fig. 7). Additionally, the change in the fraction of storms featuring rain-after-hail conditions is larger, for larger hail diameter thresholds. Therefore, for hailstorms with $\geq30\,\mathrm{mm}$ hail diameter the frequency of very intense rain-after-hail is projected to be two-fold (dotted line in Fig. 7).

While Fig. 6 shows the climate change signals to the environmental conditions of hailstorms averaged across Europe, similar analyses were performed separately for 11 European sub-regions. The results are shown in the supplementary Fig. S6 – S16 and they indicate that regional composites reveal significant differences between the sub-regions for some variables, such as CAPE, while others, like $d_{\mathrm{hail}}$, show more homogeneous signals across Europe. It is noteworthy that some regions exhibit low statistical significance, which suggests that the overall signals are not uniformly applicable across the entire simulation domain.

A similar composite analysis of thermodynamic profiles associated with storms in the current and future climate simulations reveals changes in the relevant vertical profiles (Fig. 8). The lifting condensation level (LCL) and the level of free convection (LFC) are relatively low in both climate simulations (below 1 and $2\,\mathrm{km}$, respectively), facilitating cloud formation and precipitation. The slight increase in the LCL and LFC heights may slightly hinder convection initiation, as parcels need to be lifted further (during initiation) before condensation sets in (LCL) and they reach positive buoyancy (LFC). However, this could lead to more intense and energetic storm conditions as the stronger inhibition allows for additional buildup of CAPE.

The future scenario reveals significant changes in the profiles of equivalent potential temperature, $\Theta_e$. As expected due to the warming and associated increase of specific humidity, $\Theta_e$ values are markedly higher, potentially leading to more vigorous storm dynamics (Fig. 8b). On average, $\Theta_e$ at $2\,\mathrm{m}$ in the inflow sector is projected to be $6.6\,\mathrm{K}$ higher in future hailstorms (Fig. 8c). Generally, the change in $\Theta_e$ is greatest below $600\,\mathrm{hPa}$, while the core of the storm composite also exhibits increased changes in $\Theta_e$ of about $6\,\mathrm{K}$ as opposed to $\sim4\,\mathrm{K}$ in the storm-unaffected free mid-troposphere (Fig. 8c).

Furthermore, the magnitude of the composite mean vertical wind maxima remains unchanged, however, it extends to higher altitudes (Fig. 8c). In the composite mean, the vertical wind does not scale with $w \propto \sqrt{2\,\mathrm{CAPE}}$, as the composite mean maximum vertical wind changes from $18.8$ to $18.3\,\mathrm{m\,s^{-1}}$, whereas the composite mean maximum CAPE increases from 1587

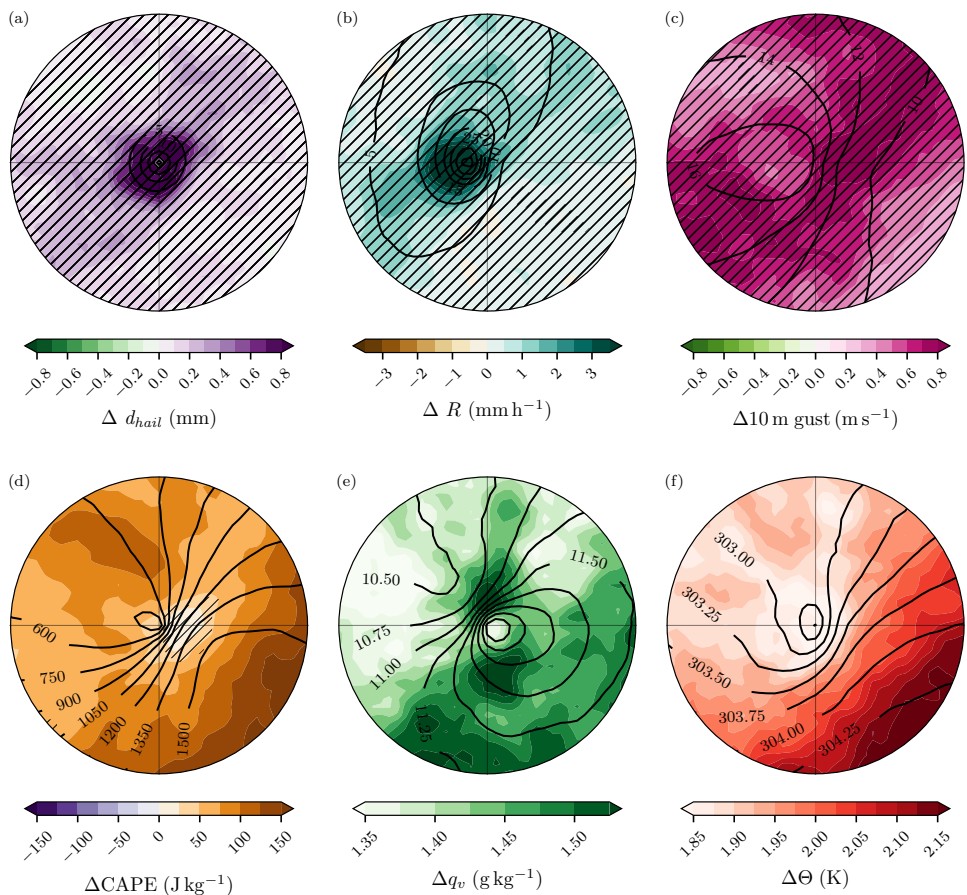

**Figure 6.** Changes (filled contours) in composite-mean horizontal fields between current (solid contours) and future climate simulations for the most intense storms in JJA ($n = 3641$ and $3284$, respectively), centered on their track center and rotated so that their respective movement vectors align with the $x$-axis (with the storm moving to the right). The radius of the figure outline is $40\,\mathrm{km}$. Shown are climate change signals in (**a**) hail diameter, (**b**) rain rate, (**c**) $10\,\mathrm{m}$ wind gust, (**d**) CAPE, (**e**) specific humidity at $850\,\mathrm{hPa}$, and (**f**) potential temperature at $850\,\mathrm{hPa}$. Hatching indicates areas with insignificant changes ($< 2\sigma$, determined through bootstrapping with $n = 1000$ resamples, see Sect. A). Versions of this figure for the PRUDENCE domains can be found in the Supplement (Fig. S6 – S16).





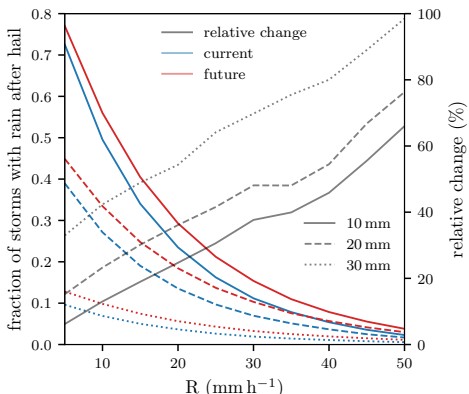

**Figure 7.** Rain after hail: Fraction of storm-centered footprints in JJA of the current (blue) and future (red) climate simulations as a function of mean rain rates ($x$-axis) in a $20{\times}20\,\mathrm{km}^2$ box behind the storm center. Grey lines indicate the relative change from current to future climate simulations. Three different storm maximum hail diameter thresholds ($10, 20, 30\,\mathrm{mm}$) are shown. The statistics include $\sim\!30\,\mathrm{k}$, $\sim\!17\,\mathrm{k}$, and $\sim\!4\,\mathrm{k}$ storm-centered footprints for $d_{\mathrm{hail}} \geq 10$ (solid), 20 (dashed), and $30\,\mathrm{mm}$ (dotted), respectively.

to $1741\,\mathrm{J\,kg^{-1}}$ from the current to the future climate. However, the sampling bias arising from the coarse vertical resolution available might lead to an underestimation of the vertical wind magnitude, partly explaining the discrepancy. As the height of the maxima changes between current and future climate simulations, this bias between vertical wind and CAPE might differ between the two[3]. The alignment of vertical wind contours with respect to ambient temperature however remains unchanged.

The storm-centered composites serve to investigate the atmospheric environment of the most intense hailstorms, revealing substantial changes for the considered future climate scenario. Key findings include increased hail diameter, intensified rainfall and wind gusts, elevated CAPE, and higher specific humidity and potential temperature (in particular in the lower troposphere). These changes suggest a warmer, moister low-level environment with enhanced energy availability, favoring more intense and hazardous storms. To further explore the response of all tracked hailstorms to changing environmental conditions, we turn to

the analysis of the storms' inflow environment.

---

[3]Looking at the hourly column maximum vertical wind, which is sampled at the model time-step ($20\,\mathrm{s}$) and includes all 60 model levels, these results remain unchanged, yielding $23.0$ and $23.3\,\mathrm{m\,s^{-1}}$ for current and future simulations respectively.



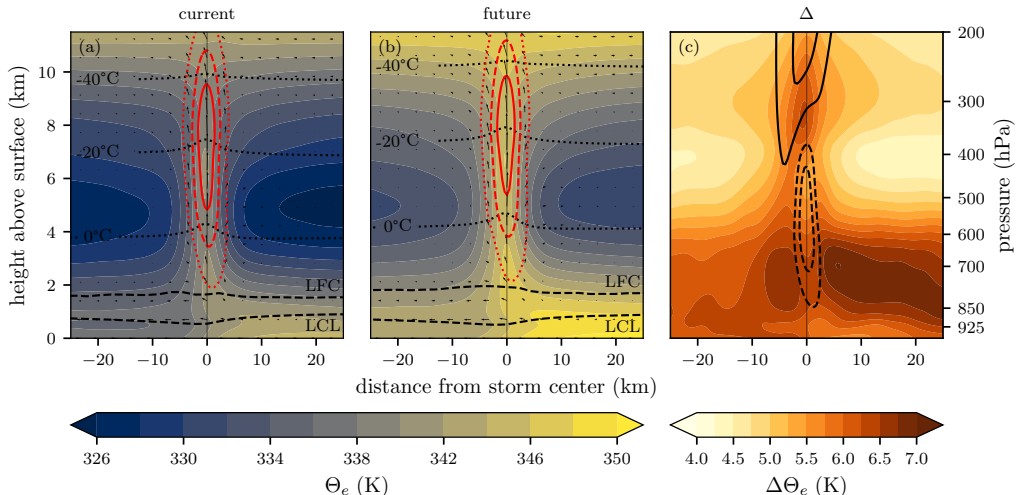

**Figure 8.** Vertical cross-section of storm-centered composites in JJA along the storms' propagation vector (storm movement to the right) in the current (**a**) and future (**b**) climate simulations, $n = 3641$ and $3284$, respectively. Filled contours show $\Theta_e$, black dotted contours the temperature $0$, $-20$, and $-40\,°C$, black dashed contours the lifting condensation level and level of free convection, and the red contours vertical wind ($5$, $10$, $15\,\mathrm{m\,s^{-1}}$, dotted, dashed and solid, respectively). (**c**) Differences (filled contours) in composite-mean fields between future minus current climate simulations, dashed and solid black contours indicate negative and positive changes in vertical wind speeds ($-1$, $-0.5$, $0.5$, and $1\,\mathrm{m\,s^{-1}}$).

## 5    Climate change effects on hailstorms' inflow environment

In this section, we first specifically investigate how CAPE, CIN, vertical wind shear, and column water vapor in the inflow sector of all tracked hailstorms change in the future climate simulation (Sect. 5.1). And secondly, in Sect. 5.2 we discuss the impact of the heightened $0\,°C$ level on the maximum hail size diagnosed with HAILCAST. Unlike in the previous Sect. 4, 340    where we analyzed the most intense hailstorms, we now include all footprints of all JJA storms in the analysis, in order to investigate the effect of the inflow environment on the response of hailstorms of different intensities.

To this end, similar to the inflow environment analysis conducted in Prein et al. (2017a), key thermodynamic variables in the storms' inflow sector ($50\times50\,\mathrm{km}^2$ box, $10\,\mathrm{km}$ ahead of the storm center, see Fig. 1) are diagnosed in the current and future climate simulations and compared to the maximum hail diameter produced by the hailstorm at a given time.



## 5.1 Change in hailstorm inflow environments

First, we consider CAPE in the inflow environment of the simulated hailstorms. In the inflow environment of hailstorms in current climate simulations, CAPE exhibits strong variability, with noticeably higher values for storms with larger hail diameters (Fig. 9a). In future climate simulations, the bulk of the CAPE distribution slightly shifts towards larger values, with similar changes for storms yielding small and large hailstones. The starkest change between CAPE in the inflow environment between

hailstorms in current and future climate simulations is present in the extreme CAPE values. There the CAPE distribution is shifted such that high-CAPE inflow environments with are more frequent in future climate hailstorms. While the mean value increases from 1814 to $1913\,\mathrm{J\,kg^{-1}}$, +5.5% (Fig. 9a), the 90th percentile of the CAPE distribution increases from 2912 to $3211\,\mathrm{J\,kg^{-1}}$, +10.3%.

Inflow maximum CIN follows a similar shift towards higher values (mean increases from 220 to $254\,\mathrm{J\,kg^{-1}}$, +15.7%),

while high-CIN environments are more prevalent in environments of $\geq 30\,\mathrm{mm}$ hailstorms compared to $\geq 10\,\mathrm{mm}$ hail storm environments (Fig. 9b). We do not find a significant reduction of relative humidity of the mean low-level inflow ($850\,\mathrm{hPa}$, not shown) as was observed by previous studies (e.g., Chen et al., 2020; Rasmussen et al., 2020). However, an increase in the moist lapse rate (see Fig. 8c) is a possible explanation for the higher CIN values in the inflow regions of hailstorms in future climate simulations.

Further, vertical wind shear ($10\,\mathrm{m}$ to $500\,\mathrm{hPa}$) in the inflow region increases, in the mean, from 14.6 to $15.5\,\mathrm{m\,s^{-1}}$, +6.0% (Fig. 9c). Further, hailstorms yielding large hail ($\geq 30\,\mathrm{mm}$) feature a distribution of vertical wind shear shifted to slightly higher values (Fig. 9c). It should be noted here, that some of the increase in vertical wind shear in inflow regions of hailstorms in current to future simulations might not only characterize the pre-storm conditions but might also be driven by the storm itself. This is because more vigorous updrafts in future climate simulations necessitate stronger low-level inflows and anvil outflows,

which in turn leads to increased vertical wind shear in the inflow sector. Further evidence of this can be found in the overlap of the inflow sampling area (see Fig.1) with the anvil (Fig.5c).

As previously mentioned in Sect. 4, increased values of CAPE are accompanied by higher amounts of water vapor in future storm environments and for storms yielding larger hail diameters (Fig. 9d). The mean total column water vapor in the hailstorm inflow environments rises from $38.5\,\mathrm{g\,m^{-2}}$ to $44.1\,\mathrm{g\,m^{-2}}$ (+14.5%), which is slightly higher than the theoretical 12.8%

that would be expected from a $1.83\,\mathrm{K}$ warmer $850\,\mathrm{hPa}$ inflow environment according to the $\sim 7\%\,\mathrm{K^{-1}}$ Clausius-Clapeyron relationship. The trend is analogous for hailstorms with $>30\,\mathrm{mm}$ hail stones (Fig. 9d).

Storm footprints that produce larger maximum hail diameters are on average associated with larger values of inflow CAPE. This relation is consistent between the current and the future simulation periods (Fig. 10a). Contrary to this linear relationship, using the hail trajectory growth model outlined in Kumjian and Lombardo (2020), Lin and Kumjian (2022) found that the

largest hail diameters tend to originate from environments with intermediate CAPE values. Notably, for any given hail diameter larger than $20\,\mathrm{mm}$, inflow maximum CAPE is significantly higher in the future climate simulations than in the current, where the opposite is true for $<15\,\mathrm{mm}$ hail (Fig. 10a).



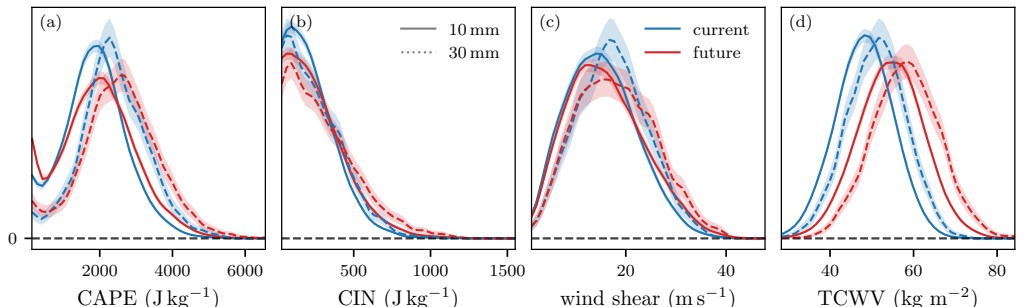

**Figure 9.** Probability density functions (scaled such that they integrate to 1) of (**a**) maximum CAPE in the inflow region, (**b**) maximum CIN in the inflow region, (**c**) mean $10\,\mathrm{m}$ to $500\,\mathrm{hPa}$ vertical wind shear in the inflow region, and (**d**) maximum total column water vapor in the inflow region (Fig. 1) of hailstorms in current and future climate simulations. The distribution is determined based on $\sim\!30\,\mathrm{k}$ and $\sim\!4\,\mathrm{k}$ storm-centered footprints for $d_{\mathrm{hail}} \geq 10$ (solid) and $30\,\mathrm{mm}$ (dotted) respectively. Lines show the mean, while the shaded area denotes the 5–95$^{\mathrm{th}}$ percentile range of $n = 1000$ bootstrapped samples.

### 5.2 Change in freezing level height

Previous studies have emphasized the role of increased melting due to a higher $0\,^{\circ}\mathrm{C}$ level height (e.g., Gensini et al., 2024),
particularly for smaller-diameter hailstones which are expected to experience strongest melting due to their slower fall velocity. To examine this effect in our simulations, we first analyze the relationship between hailstorm-environmental freezing level heights, resulting hail diameters, and the vertical wind speeds required to support hailstones of different sizes. To this end, we compare $0\,^{\circ}\mathrm{C}$ level heights, instantaneous storm-maximum hailstone diameters, and vertical wind speeds across current and future climate simulations.

Storm footprints associated with larger storm maximum hail diameters are generally linked to higher mean $0\,^{\circ}\mathrm{C}$ level heights in the inflow. This relationship remains consistent across both the current and future simulation periods (Fig. 10b). In inflow environments of hailstorms in future climate simulation, the $0\,^{\circ}\mathrm{C}$ level rises by $362\,\mathrm{m}$ compared to current climate storms ($+9\%$, Fig. 10b). Results show that the $0\,^{\circ}\mathrm{C}$ level height is consistently higher in future hailstorms, regardless of maximum hail diameter (Fig. 10b). This change in $0\,^{\circ}\mathrm{C}$ level is accompanied by an increase in the level of homogeneous freezing (i.e.,
isotherm of $-38\,^{\circ}\mathrm{C}$) of $481\,\mathrm{m}$ ($+5\%$), which leads to a 2.3% increase in hail growth layer thickness. This increase allows hailstones slightly more time in the updraft to grow to larger diameters. However, a higher $0\,^{\circ}\mathrm{C}$ level also extends the time hailstones spend in above-freezing air during their descent, enhancing melting and potentially reducing their final diameters. Interestingly, storms producing the largest hailstones are associated with higher $0\,^{\circ}\mathrm{C}$ levels in both present and future climates. If $0\,^{\circ}\mathrm{C}$ level height played a dominant role in determining maximum hail sizes, an inverse relationship would be expected —
yet no such trend is observed, suggesting that melting has a limited effect on the storm maximum hail diameters.



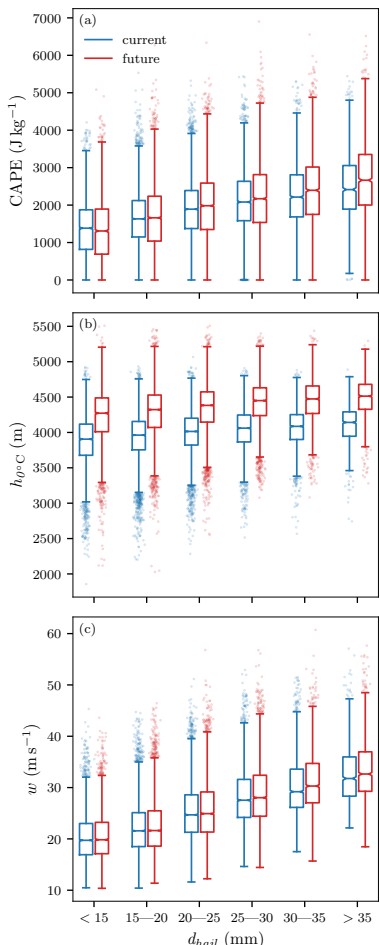

**Figure 10.** Statistics of storm characteristics for different bins of momentary storm maximum hail diameters (5 mm wide bins) in current and future simulations ($n = 31\,899$ and $28\,259$ storm centered footprints respectively). Characteristics shown are (**a**) inflow maximum CAPE, (**b**) mean $0\,°C$ level height calculated in the inflow region, and (**c**) maximum vertical wind within the storm during the last hour (sampled at the 60 model vertical levels and 20 s model time-step). The horizontal line with notches shows the mean and confidence interval respectively, while the box covers the interquartile range (IQR) and the whiskers extend to the farthest data point lying within $1.5\times$ the IQR from the box. Outliers (points) are jittered horizontally to increase legibility.



Increased melting of hailstones in a future climate (through an increased $0\,°C$ level height) could be compensated by stronger updrafts leading to increased hail growth. To address this compensating effect, we analyze the relationship between hail diameters and vertical wind speeds. As expected, in both current and future climate simulations we observe a positive relationship between storm maximum vertical winds and storm maximum hail (Fig. 10c). Storms with hail diameters exceeding $25\,mm$

exhibit a slight (+3%), significant increase in the maximum vertical wind in future climate simulations compared to current climate simulations (Fig. 10c), but no such trend is present for storms yielding only small hailstones. This further reinforces the conclusion that hailstone melting exerts a limited influence on storm maximum hail diameters in the simulated hailstorms.

On the other hand, melting likely plays a key role in the reduced hail swath areas below $11\,mm$ in future climate simulations as discussed in Sect. 3 (Fig. 4b). Conceivably, melting also influences hail diameters above that, however, those changes are

outweighed by more favorable convective environments in future climate simulations. The local maximum in the fractional change in the swath area curve at $15\,mm$ is a probable consequence of a crossover from increased melting being relevant at smaller diameters and less so at larger diameters (Fig. 4b).

In summary, the projected increases in CAPE and specific humidity due to a warming climate emerge as key drivers of enhanced hailstorm intensity in future scenarios. In contrast, the anticipated increase in hailstone melting due to elevated $0\,°C$

level heights is not relevant for the storm maximum hail diameters and only affects the extent of the smallest hail diameters. These findings underscore the dominant role of thermodynamic intensification in shaping future hailstorm environments.

## 6   Summary and conclusions

This study highlights the significant impacts of global warming on hailstorm tracks and dynamics across Europe, demonstrating that future climatic conditions are likely to result in more intense, larger, and regionally more frequent hailstorms. Using the

output from the COSMO v6 model simulations (Cui et al., 2024) with a $+3\,K$ PGW approach performed by Thurnherr et al. (2025), we conducted additional hailstorm-tracking and object-based analyses. They allowed us to capture projected changes in hailstorm structure and environment, providing valuable information for understanding regional hazards.

We can now address the questions posed in the introduction;

1. Storm area and lifetimes are projected to increase by +10.3% and +1.7% respectively across the simulation domain.
The change in storm area features extended areas of positive (and significant change), while the lifetime shows a strong spatial variability, with only some extended areas in eastern France, Germany, and eastern Europe exhibiting coherent positive trends (Fig. 3c,f).

2. Further, the mean storm propagation velocity increases by +7.3%. Most regions show significant positive changes of $+1$ to $+3\,m\,s^{-1}$, suggesting faster-moving hailstorms under future warming (Fig. 3i). Climate change influences the spatial
extent of hailstorms by modifying per-storm hail swath areas and altering hailstorm frequency.



3. Our analysis shows a decrease in the frequency of storms producing hail smaller than $\leq 25\,\mathrm{mm}$, while larger hail events ($\geq 45\,\mathrm{mm}$) become more frequent, with a two-fold increase expected for hail above $\sim 45\,\mathrm{mm}$. Per-storm hail swath areas increase for most hail sizes, with the most significant increase observed for $15\,\mathrm{mm}$ hail ($+33\%$), and when considering both frequency and swath area changes, swath area expansion dominates for hail below $40\,\mathrm{mm}$, whereas frequency changes are more influential for larger hail (Fig. 4a,b).

4. The intensification of hailstorm activity is closely associated with enhanced potential for larger updrafts, as evidenced by elevated CAPE values and increased specific humidity levels, both of which are driven by the higher moisture content associated with rising temperatures. These increases in specific humidity and CAPE values are consistent with the mechanisms discussed by Prein et al. (2017a) and Trapp et al. (2019), who emphasized the role of convective potential in the development of convective storms and hail formation under future climate scenarios. Contrary to expectations (i.e. Dessens et al., 2015), the higher $0\,^{\circ}\mathrm{C}$ level heights resulting from climate change did not significantly impact storm maximum hailstone diameters, suggesting that factors such as more volatile convective environments and resulting increased updraft strength may mitigate potential melting effects. In our analysis, the effect of increased melting is constrained to the smallest hail diameters ($\leq 11\,\mathrm{mm}$). This analysis contributes additional nuance to prior studies, aligning with the uncertainties highlighted by Mallinson et al. (2023) and Battaglioli et al. (2023).

5. As discussed in the introduction, previous studies have suggested an increase in hailstorm frequency and intensity under warmer conditions. Our results corroborate these trends, demonstrating regionally diverse trends in hailstorm frequency and an increase in the storm maximum hail diameter across Europe. These findings align with projections from studies such as Raupach et al. (2021), Wilhelm et al. (2024), Thurnherr et al. (2025), and Feldmann (2025), which identified similar regional trends. An important result of this study is that the hail environments change similarly in all sub-regions — regardless of the local hailstorm frequency trend signal. Notably, CAPE in the inflow increases in all sub-regions, even in regions where the Eulerian summer mean CAPE decreases (Thurnherr et al., 2025). This distinction highlights that it is not primarily the hail events themselves that are changing, but rather the frequency of favorable environmental conditions for the initiation of severe convective storms.

6. Hailstorms under future climate conditions remain potent threats due to accompanying changes in both precipitation intensity and near-surface wind. A comprehensive analysis of storm parameters reveals that rain rates around the storm center increase by $+20.0\%$ while $10\,\mathrm{m}$ wind gusts intensify by $+5.1\%$ (Fig. 6b,c). Concurrently, CAPE and CIN shift toward higher values $+5.5\%$ and $+15.7\%$ respectively (Fig. 9a,b), and vertical wind shear strengthens ($+6.0\%$, Fig. 9c). Collectively, these factors point to more pronounced localized rainfall, stronger near-surface winds, and a more unstable convective environment — conditions that could foster not only larger hail but also compound hazards such as wind-driven hail or intense rainfall immediately following hail, where the latter is projected to increase by more than 50% (Fig. 7).




Despite the robust insights provided by this study, several limitations merit consideration. The resolution of the COSMO v6 model ($2.2\,\mathrm{km}$), being convection-permitting, not convection-resolving, although suitable for regional climate simulations,
may not fully capture the fine-scale processes influencing hailstorm development (Bryan et al., 2003). The HAILCAST model relies on simplified parameterizations that may not fully represent the complex interactions between microphysical processes and storm dynamics. Two-moment microphysics schemes, which predict both mass and number concentration of hydrometeors, could offer a more accurate depiction of hail formation and evolution and would also allow for moisture tagging experiments. Additionally, a full 3D hail trajectory analysis could further refine the model's ability to simulate hailstorms by explicitly
tracking hailstones' paths through a storm. However, these advanced approaches are computationally intensive and are currently impractical for long-term climate simulations.

As hailstorms locally become more common and intense, the demands on insurance systems to manage the financial risks associated with hail-related damages will escalate. Developing comprehensive risk assessment frameworks that incorporate climate projections and socioeconomic factors will be essential for informed decision-making and effective resource allocation.
To this end, the comprehensive dataset of hail swaths in current and future climate simulations ($2\times \sim 40\,\mathrm{k}$) provided in the *Code and data availability* section is well suited for a stochastic resampling approach and application as a hazard in a hail damage model.

*Code and data availability.* The tracking algorithm (Brennan et al., 2024) used in this study is available under https://doi.org/10.5281/zenodo.12685276. The script used to extract the storm environment and construct composites is available at https://doi.org/10.5281/zenodo.
14631622. Storm track data, storm-centered footprints, and the hail swath dataset will be made available through the ETH research collection https://www.research-collection.ethz.ch during the review process. The comprehensive dataset of hail swaths is well suited for a stochastic resampling approach and application as a hazard in a hail damage model. It includes per-storm areas with corresponding HAILCAST maximum hail diameters and is available for the current and future simulations with 39 594 and 37 746 swaths, respectively. The compressed dataset amasses to a total of ~0.7 GB. As outlined by Brennan et al. (2024), gap-filling techniques were employed to mitigate the "fishbone
effect", a term introduced by Lukach et al. (2017). This phenomenon refers to the fragmented hail swaths resulting from the limited temporal resolution when capturing fast-moving storms with short paths along their direction of motion. The effect poses a particular challenge for damage models, as it leads to an underestimation of the hail-affected area, especially for larger hail diameters.

## Appendix A: Bootstrapping

In order to visualize the object-based characteristics resulting from the storm tracking on a map (Fig. 2 and 3), the track
data was gridded using a circular disk kernel with $r = 50\,\mathrm{km}$ ($7\,854\,\mathrm{km}^2$). For the evaluation of statistical significance, block bootstrapping was utilized. To this end, the simulated years were randomly pooled with replacement from the original sample until the same number of years as in the original dataset was obtained. This resampling was repeated $n = 1000$ times for



both the current and future climate periods. Statistical significance ($2\sigma$) was determined by taking the difference between each resampled mean pair and determining wether 0 lies outside of the 5 to 95% interval of the distribution of differences. The same

approach was applied to the composite differences (Fig. 6).

*Author contributions.* During the extensive development of this study, all co-authors contributed valuable comments and suggestions through in-depth discussions. Furthermore, specific contributions include **KB**: conceptualization of the study, development of the methodology and data analysis, visualizations and manuscript writing, review, and editing. **IT**: computation of the simulations, review, and editing. **MS**: methodology, review, and editing. **HW**: conceptualization, review, and editing.

*Competing interests.* The authors declare no conflict of interest.

*Acknowledgements.* We extend our gratitude to our colleagues from ETH, and the entire scClim team (https://scclim.ethz.ch/) for their valuable contributions and discussions. This study was funded by the Swiss National Science Foundation (SNSF) Sinergia grant `CRSII5_201792`. We acknowledge the use of OpenAI's GPT in assisting with language refinement in the preparation of this manuscript.



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
