# Peer review of "Insights from hailstorm track analysis in European climate change simulations"

_EGUsphere, 2025_

## Author Comment (AC1)

**Final author comments for paper egusphere-2025-918**

**Insights from hailstorm track analysis in European climate change simulations**

by Killian P. Brennan, Iris Thurnherr, Michael Sprenger, and Heini Wernli

June 23, 2025
* * *
**Reviewer 1**

**Overview**

This is an excellent article embedded in a series of articles based on convection-allowing simulations in a current and a warmer climate. This particular study nicely analyzes individual storm properties, mainly showing that hailstorms will produce more severe hail due to stronger updrafts, that the melting of hail does not have a large effect on the hail size in a given storm, and that reduced frequency of hail in some parts of Europe is likely attributed to a reduced frequency of storms and not a reduction in severity. These results are highly relevant.

The figures and scientific language are of high quality and the manuscript has a clear structure. I only have one general major comment. However, since this might affect the robustness of the whole methodology and all studies using these simulations, I recommend major revisions.

**Reply**: We thank the reviewer for their very positive and constructive review and appreciate the recognition of the scientific significance of our study. Below, we provide detailed responses to the specific comments and outline the revisions we have made to improve the clarity and presentation of our study.

**General comments**

**Reviewer Comment 1.1** — My main concern is that the simulated storms are insufficiently resolved to represent real hailstorms, so the study might yield (partially) misleading results. It has been shown that at $\approx 2\,\mathrm{km}$ resolution as used here, peak updraft speeds are substantially reduced (e.g., Adlerman and Droegemeier 2002 and references therein). This issue might also affect other updraft characteristics like width and depth. Since these factors are extremely important for hail growth (e.g., Lin and Kumjian 2022 and references therein) a strong impact can be expected. I

think Hailcast somewhat corrects for these factors (Adams-Selin and Ziegler 2016) but it is unclear how well it performs with the COSMO model and at this particular resolution.

How severe these limitations are is not discussed in the article and is thus difficult to say. However, there are two reasons that at least point towards a strong bias in the simulations.

(**a**) The sister-study that goes more into the verification of the model (Cui et al. 2024) shows that the hail distribution in the tested cases is not well covered. As far as I know, this preprint is still under review, but I'm not a reviewer on it so I don't know if it will be accepted. Since the methodology is the basis for all follow-up articles, like the present one, I'm somewhat hesitant to recommend publications, at least not without emphasizing these limitations.

(**b**) Lines 323-326: the obvious discrepancy between CAPE and the maximum $w$ in the simulations is not discussed. For example, Fig. 10 at least roughly suggests that average CAPE values of for example $1500 \, \mathrm{J \, kg^{-1}}$ leads to maximum updrafts of $20 \, \mathrm{m \, s^{-2}}$ in your simulations, while from the $w_{max}$ equation (line 324) one would expect $55 \, \mathrm{m \, s^{-2}}$. Yes, the latter is a theoretic value not necessarily realized due to entrainment, but especially hailstorms (often supercells) can realize most of their CAPE (e.g., Peters et al. 2019, 2020a). Also, vertical velocities $> 50 \, \mathrm{m \, s^{-2}}$ are likely common in supercells (e.g., Peters et al. 2020b) while only extreme outliers reach this range in your simulations (your Fig. 10). This is consistent with the expected negative bias in updraft intensity at $2.2 \, \mathrm{km}$ resolution (Adlerman and Droegemeier 2002).

So far, the authors only briefly discuss that "fine-scale processes influencing hailstorm development" might not be represented (line 460). I think this insufficiently describes the problems of the simulations for the reasons outlined above. A much deeper discussion is necessary and the reader should be made aware which results must be taken with a grain of salt because of the biases in updraft characteristics (see, e.g., comment 12 below, but I could see that there are other less obvious implications).

I think it is still worth to publish these important results but to me there is some uncertainty to all of them, which must be made clear.

**Reply 1.1**: We thank the reviewer for raising the important issue of resolution-related uncertainties in simulating deep convective processes relevant for hail formation. We fully agree that kilometre-scale models, including the $2.2 \, \mathrm{km}$ COSMO simulations used here, operate within the so-called "grey zone" where small-scale turbulence and convective dynamics are only partially resolved. Nevertheless, this resolution represents the current state of the art for convection-permitting climate simulations over continental domains and has been widely used in recent studies (e.g., Prein et al., 2021). In particular, Prein et al. (2021) demonstrated that while some characteristics (such as updraft core structure and vertical mass transport) continue to improve with grid spacing $< 2 \, \mathrm{km}$, most climate change signals in mesoscale convective systems (MCSs) are already robustly captured at $4 \, \mathrm{km}$ grid spacing compared to reference large-eddy simulations with a grid spacing of $250 \, \mathrm{m}$.

Currently, a further refinement to sub-kilometre grid spacings remains computationally unfeasible for multi-year simulations over large domains. The resolution-related limitations and associated impacts on hail-relevant processes are discussed in lines 458 and following; we agree that this could be clarified further and have added appropriate text passages (see addition **b** below).

Concerning point (**a**): We mention that the accompanying study (Cui et al., 2024), which provides a dedicated verification of the same modeling framework, was accepted on 31 May for publication in J. Geophys. Res.—Atmospheres. Concerning point (**b**): We acknowledge the discrepancy between CAPE and modeled updrafts and will elaborate on its implications in the revised version, including selected references you suggested.

Manuscript addition – (**a**) reference to Cui et al. (2024), insertion on L96: "Despite limitations associated with the native grid resolution, this validation indicates that the overall spatial and seasonal patterns of hail activity are reasonably well captured."

Manuscript addition – (**b**) discussion of CAPE vs. $w$ discrepancy, addition on L326: "This underestimation is a known feature of kilometre-scale models and has been documented in several studies (e.g., Adlerman and Droegemeier, 2002; Peters et al., 2020a). It results from coarse resolution entrainment, insufficient core thermal contrast, and smoothing of updrafts. While HAILCAST attempts to account for subgrid updraft structure (Adams-Selin and Ziegler, 2016), the magnitude of this compensation at 2.2 km grid spacing remains uncertain and is an important limitation when interpreting simulated absolute hail sizes."

Revised discussion in paragraph on limitations (insertion on L460): "The simulation of convective storms at 2.2 km grid spacing inherently limits the representation of fine-scale processes critical for hail formation, such as peak updraft speed, entrainment, and hydrometeor interactions within the convective core. As discussed in Prein et al. (2021), even though kilometre-scale models significantly improve upon coarser-grid simulations and reproduce robust climate signals in convective systems, they operate in the "grey zone" where vertical mass fluxes and core dynamics are underrepresented. This likely contributes to underestimated vertical velocities (see also Fig. 10), as compared to theoretical expectations or observations in supercells (Peters et al., 2020b,a). Consequently, hail growth potential may be suppressed in the model compared to reality, particularly for severe events. While HAILCAST includes parameterizations to adjust for updraft strength (Adams-Selin and Ziegler, 2016), uncertainties remain regarding its performance at this specific resolution and model setup. Acknowledging this, we caution that the absolute hail sizes should be interpreted in a relative sense, with a focus on comparative patterns rather than absolute magnitudes."

**Specific comments**

**Reviewer Comment 1.2** — Lines 32-33: can you add a reference to your statement that CC scaling leads to stronger average updrafts and hail? I agree that it is so on average but as you mention further below these links are not 100% clear, so at least providing a reference here seems warranted.

**Reply 1.2**: Thank you for pointing this out, we've added the appropriate reference (Raupach et al., 2021).

**Reviewer Comment 1.3** — Line 33: Also here you could add a reference showing the rise in the 0°C level (e.g., Prein and Heymsfield 2020, Gensini et al. 2024).

**Reply 1.3**: We've added the suggested references (Prein and Heymsfield, 2020; Gensini et al., 2024).

**Reviewer Comment 1.4** — Line 35: Is the slower fall speed really the main reason why small hail is more affected from melting? I always thought it is that larger hail has a relatively smaller surface area and more mass, so the cooling from latent heat exchange can slow down melting more effectively. Can you perhaps add a reference?

**Reply 1.4**: Appropriate references added (Pruppacher and Klett, 2010; Raupach et al., 2021).

**Reviewer Comment 1.5** — Line 46: Several other means come to mind: observational proxies like lightning and overshooting tops (e.g., Punge and Kunz 2017), hail pads (e.g., Manzato et al. 2022), hail reports and radiosonde soundings. You don't need to mention all possible methods though. Perhaps just rephrase that the means you introduce here are the ones that have so far been used in the literature to study climate trends (perhaps adding hail pad studies).

**Reply 1.5**: Thank you for pointing this out, an appropriate footnote has been added on L46: "Please note that this list is non-exhaustive."

**Reviewer Comment 1.6** — Lines 55-56: This might be confusing to some readers. The European domain is not larger than the US, no? I suggest rephrasing.

**Reply 1.6**: We thank the reviewer for this input and have omitted that phrase (L55–56) from the manuscript.

**Reviewer Comment 1.7** — Line 173: It is not immediately clear what "mean storm maximum hail diameter" is. Is it the average over all maximum diameters occurring within the storms at a gridpoint? Consider defining it more clearly once.

**Reply 1.7**: Thank you for pointing out this ambiguity, we've rephrased the sentence on L173 to: "Storm maximum hail diameters are on average in the range of 18 to 25 mm throughout Europe."

**Reviewer Comment 1.8** — Line 179: I'm assuming this refers to the area that is over the 10 mm threshold? Consider mentioning this explicitly since area could also refer to other variables like precipitation.

**Reply 1.8**: We've added more details, the sentence now reads: "Furthermore, mean instantaneous storm areas (as defined by the area that exceeds 10 mm hail diameter, see Sect. 2.3 for details)..."

**Reviewer Comment 1.9** — Line 257: Consider replacing "drawn in" with "generated", which seems more accurate.

**Reply 1.9**: Changes made according to suggestion.

**Reviewer Comment 1.10** — Line 261: Can you add a reference or some more context for why you consider this "the inflow level"? Can't the inflow be dominated by any layers in the lowest 3 km or so and vary substantially for example between elevated storms and supercells (e.g., Nowotarski et al. 2020)?

**Reply 1.10**: Thank you for pointing this out, we've added the following footnote on L261: "Although the inflow level can vary substantially (e.g., Nowotarski et al., 2020), due to limited data availability (the simulation output for 3D variables was only available on 8 vertical pressure levels) and prior knowledge of inflow characteristics of hailstorms in the alpine domain (Brennan et al., 2024), 850 hPa was selected as the inflow level."

**Reviewer Comment 1.11** — Lines 326-327: It's not clear to me what you mean by "sampling bias due to coarse vertical resolution". Aren't all model grid points used? Then the $w_{max}$ in the

simulated cell is what matters for the simulated hail production. So in other words, it's not a sampling bias but more a model bias which might significantly impact the whole methodology (see general comment 1). Or did I misunderstand something?

**Reply 1.11**: Yes, all model grid points are used, but for this analysis the vertical wind speed is only available on 8 pressure levels and not on the model grid. If the vertical wind maxima changes in its altitude (see Fig. 8) and shifts from falling onto a vertical level to falling between the pressure levels (or vice-versa) then this would result in a bias. No changes were made to the manuscript regarding this comment.

**Reviewer Comment 1.12** — Lines 362-364: Agreed! One way to test this would be to repeat the analysis for the far-field environment. The appropriate distance from the storm could be determined based on where you stop seeing storm-induced perturbations in the pressure and wind field (Fig. 5) and on the literature (Coniglio and Parker 2020; in general this and other references within could be added to this paragraph). Would this feasible?

**Reply 1.12**: From the Eulerian analyses of the same climate simulations in Thurnherr et al. (2025), the seasonal (JJA) average increase of 0–6 km shear over land in Europe is around $1 \, \mathrm{m \, s^{-1}}$. One could argue that, since the value given in Thurnherr et al. (2025) (revised version) corresponds to a seasonal average and convective storms occur relatively rarely, this would come close to an assessment of the far-field environment. The fact that this Eulerian change in vertical wind shear is very close to the changes in storm-inflow vertical wind shear we determine in our analysis, indicates that the bulk of the changes in the storm-inflow vertical wind shear are likely attributable to environmental changes, rather than self-driven. Determining the far-field effects in an object-based framework would require significant additional analysis, putting such an investigation beyond the scope of this manuscript.

**Reviewer Comment 1.13** — Lines 372-376: The underestimation of maximum updraft intensity outlined in general comment 1 could be the reason why no nonlinear behavior is seen in your simulations compared to Lin and Kumjian.

**Reply 1.13**: Please see Reply 1.1.

**Reviewer Comment 1.14** — Line 380: Here or later in the discussion it might be helpful to clarify what exactly "small" refers to. Gensini et al. argue that melting dominates for hail < 4 cm (their Fig. 1) while in your study "small" is used for much smaller diameters (e.g., lines 439, 426). I think this clarification is important to put your work into perspective because it emphasizes that your results point in a different direction.

**Reply 1.14**: Thank you for pointing out this discrepcancy. We've adressed this comment by specifying the diameter on L380, the sentence now reads: "Our analysis showed that the frequency of large hail is increasing in the hail swaths in a warmer climate (Fig. **??**). Changes in hail size distributions due to a warmer climate have previously been attributed to increased updraft speed leading to larger hail sizes aloft, and a higher 0°C level height, particularly affecting smaller-diameter hailstones (e.g., < 40 mm, Gensini et al., 2024), which are expected to experience the strongest melting due to their slower fall velocity and small mass (Brimelow et al., 2017)." Furthermore, changes outlined in Reply 1.15 address the discrepancy further.

**Reviewer Comment 1.15** — Section 5.2: I've never used HAILCAST but if I recall correctly, accurately representing melting of hail in such a model is not an easy task and subject to high uncertainty. If you agree, this point should be discussed, because it could be the reason why melting does not have a strong effect in your simulations compared to the other literature.

**Reply 1.15**: We agree that accurately representing hail melting is challenging and subject to substantial uncertainty in HAILCAST. The model estimates predominantly the maximum hail size and applies a simplified melting parameterization. This limitation could explain why our results show a weaker sensitivity to melting level height compared to studies that consider the full hail size spectrum (e.g., Gensini et al., 2024). We have revised the manuscript to explicitly acknowledge this uncertainty in Section 5.2 and added the following paragraph to L407: "It should be noted here that the results concerning melting are subject to uncertainty due to the simplified representation of melting processes in HAILCAST, which likely lacks the complexity required to, for example, accurately capture hailstone ablation during descent. Furthermore, HAILCAST predominantly estimates the maximum hail size and does not resolve the full size spectrum; potential melting effects on smaller hailstones may be underestimated, limiting the model's sensitivity to variations in melting level height. This provides a possible explanation for the discrepancy between results based on Thurnherr et al. (2025) and those presented in Gensini et al. (2024). Thurnherr et al. (2025) see a similar effect as Gensini et al. (2024), but the transition from decreasing to increasing hail occurrence occurs at 12.5 mm in the former analysis, similar to our object-based analysis, while the latter threshold lies around 40 mm. This difference likely arises from methodological differences — these studies use different hail parameterizations, which result in distinct hail size distributions, and may also be influenced by model resolution and storm dynamics in the respective simulations. For further discussion, see Thurnherr et al. (2025). Further, our object-based approach focuses on long-lived storms, which tend to produce larger hail, potentially excluding the smaller sizes most affected by melting. Moreover, if future storms yield more large hailstones due to stronger updrafts, increases in melting level height may become less relevant for the largest hailstones."

Please refer to the track changes document regarding additional changes that have been made to the manuscript regarding this comment and Reviewer Comment 1.14.

**Reviewer Comment 1.16** — Line 424: The last sentence here seems out of place and could be removed?

**Reply 1.16**: Removed as suggested.

**Reviewer Comment 1.17** — Line 440: I'd suggest briefly mentioning what "uncertainties" you are referring to because the references alone leave room for interpretation.

**Reply 1.17**: Thank you for pointing out the lack of precision here, the corresponding sentence now reads: "This analysis contributes additional nuance to prior studies (e.g., Battaglioli et al., 2023), aligning with the uncertainties originating from how microphysical processes are represented in numerical models highlighted by Mallinson et al. (2023)."

**Reviewer Comment 1.18** — Lines 447-449 and 456: I fully agree and these are important results.

**Reply 1.18**: Thank you for this positive comment!

**Reviewer Comment 1.19** — Line 460: I recommend also citing Adams-Selin (2025) here.

**Reply 1.19**: We've added a reference to Adams-Selin (2025).

**Technical corrections**

**Reviewer Comment 1.20** — Line 153: So you mean "SON" for November?

**Reply 1.20**: Of course, thanks for bringing this to our attention.

**Reviewer Comment 1.21** — Line 204: Consider replacing "different" with "spatially heterogeneous" and "similar" with "homogeneous" to be clearer.

**Reply 1.21**: Thank you for these suggestions which we've implemented in the revised version.

**Reviewer Comment 1.22** — Footnote 2: Consider shortening to "The second approach is arguably better as..."

**Reply 1.22**: We believe the extended length of this footnote is justified by the reduction in ambiguity it provides thus opting to not abbreviate it.

**Reviewer 2**

**Summary**

This is a very interesting study looking at numerous aspects of hail in Europe under a warming climate (e.g. number of hailstorms, hail diameter and swath size). They nicely couple this with the inflow environment of the storms again in the current/future climate framework. The study is suitable for the journal and shows several novel findings, but the analysis could be adapted further, and some methods/datasets are based on unpublished papers. Therefore, I recommend major revisions before publishing.

**Reply**: We thank the reviewer for the positive and encouraging assessment of our study and are pleased that they find the topic, scope, and results to be novel and well-suited for the journal. We appreciate the recognition of our efforts to comprehensively analyze hailstorm characteristics—including hail size and swath area—in the context of environmental inflow conditions under both present and future climates.

   We acknowledge the reviewer's concerns regarding the reliance on methods and datasets described in currently unpublished companion studies. In the meantime the study by Cui et al. was accepted for publication in J. Geophys. Res., and by the the time this manuscript goes into the next iteration, the companion study by Thurnherr et al. is expected to be accepted as well.

   We have carefully considered and incorporated the reviewer's suggestions and are confident that, as a result, the revised manuscript is significantly improved in clarity, rigor, and accessibility.

**Main comments**

**Reviewer Comment 2.1** — In my view the authors try a bit too hard to match their findings with previous studies and cite some papers that are not available or unreviewed preprints. I'd suggest that they stick to primarily citing current peer-reviewed literature. The authors could also consider delaying the publishing of the manuscript until there are further papers available to support their conclusions and the datasets they use.

**Reply 2.1**: We appreciate the reviewer's concern regarding the citation of preprints and unpublished work. We agree that peer-reviewed literature should form the backbone of any scientific discussion.

   At the same time, we note that some of the methods and datasets employed in this study are part of an ongoing coordinated research effort. We will update the citations accordingly as soon as the companion studies have been accepted (Cui et al. was accepted on 31 May; the revised version of Thurnherr et al. is under review). We aim to communicate these results in a timely manner while acknowledging the evolving nature of the supporting literature.

**Reviewer Comment 2.2** — The authors mention the possible changes in convective initiation and how that may affect hailstorm frequency (e.g. L203-204). I think this is a plausible hypothesis

and should be investigated in more detail. For example, the authors could set a reflectivity threshold as a proxy for convective initiation (CI) e.g. 40 dBZ and look at changes in CI frequency in Europe between current and future. Maybe this will show similarities to the plot shown in Fig. 2c.

**Reply 2.2**:  We thank the reviewer for this thoughtful and constructive suggestion. We agree that changes in CI may play an important role in modulating future hailstorm frequency and that investigating this aspect in more detail could strengthen the interpretation of our results.

Unfortunately the simulation output we have available as a basis for our study does not include synthetic radar reflectivity and also does not allow calculating reflectivity offline, thus your suggestion falls outside of the possibilities of our study. We refer to Thurnherr et al. (2025, revised version), where a seasonal mean increase in CIN is seen in regions with a decrease in hailstorm frequency.

**Reviewer Comment 2.3**  —  I also think they could do more to consider kinematics, i.e. considering hail path trajectories, which is key in determining the hail size on the ground.

**Reply 2.3**:  We thank the reviewer for highlighting the importance of storm kinematics and hail path trajectories in determining hail size at the surface. We agree that incorporating kinematic aspects, such as storm-relative motion and the trajectory of hail embryos through the updraft and downdraft regions, could provide deeper insight into hail growth and fallout.

However, as HAILCAST is only a one-dimensional hail growth model, the representation of the kinematics and therefore the feasible analysis thereof is very limited in our simulations, rendering the corresponding additional analysis outside the scope of our study. We realize that in this and the previous comment we unfortunately have to classify two very good suggestions as unfeasible. Please note that these 10 year simulations, given the high resolution and large model domain, produced enormous output and were computationally only possible with a simplified hail growth model like HAILCAST. In other words, the ambition to produce continental-domain kilometer-scale climate simulations comes at the prize of limiting the depth of the possible diagnostic.

**Inline comments**

**Reviewer Comment 2.4**  —  L30-38: Whether convective initiation will become more or less frequent should already be mentioned in the introduction.

**Reply 2.4**:  Thank you for this input, we've added the following sentence on L39: "Lastly, thunderstorm initiation is also highly sensitive to small changes in low-level temperature and moisture; however, the effect of climate change on the frequency of convective initiation is less clear (Raupach et al., 2021)."

**Reviewer Comment 2.5**  —  L70: Which simulations are the authors talking about? Please be more specific which simulations are being used, on the previous lines both Cui et al. (2024) and Thurnherr et al. (2025) are cited.

**Reply 2.5**:  Thank you for pointing out this ambiguity, the paragraph now starts with: "The Thurnherr et al. (2025) simulations apply..."

**Reviewer Comment 2.6** — Both Cui et al. (2024) and Thurnherr et al. (2025) are preprints. More specifics of the simulations in the methods and data section should be included if this study is to be published first.

**Reply 2.6**: Since we already sent back the page proofs for Cui et al., this paper will appear soon and therefore can serve as the solid basis for the description of the simulations.

**Reviewer Comment 2.7** — L124-141: It would be useful to include information about the time component. Do the hailstones just need to meet these criteria for one timestep (i.e. 5 minutes)?

**Reply 2.7**: Thank you for pointing this out, we've included this detail at the end of the paragraph: "Features must exist for at least 15 min in order to be be included in the tracks."

**Reviewer Comment 2.8** — The hailstorm definition used for tracking could also be better summarized in a table.

**Reply 2.8**: Thank you for the suggestion. We believe the current description provides a concise and self-contained explanation of the hailstorm definition. Adding a table would not significantly enhance clarity or brevity in this case.

**Reviewer Comment 2.9** — L169: RE: reference to Feldmann (2025). I'm generally ok with authors citing personal correspondence if there is a good reason to do so such as communication with a forecaster about model biases because the forecaster will not have any publications. However, just a simple citation with no further details should not be included. Maybe the person has an upcoming publication on the topic, however no paper is given in the reference.

**Reply 2.9**: A pre-print is now available for this publication (Feldmann et al., 2025), and will be used as reference instead of the personal correspondence.

**Reviewer Comment 2.10** — L202: By "hail track frequency" I am assuming the authors are referring to Figures 2a,b,c which is labeled as "n storms per season". I'd suggest using consistent terminology e.g. n hailstorms per season and use the term hailstorm throughout. In section 2.3 it is made clear that your hailstorm definition is based on the tracking algorithm.

**Reply 2.10**: Thank you for pointing this out we've replaced the only instance of "hail track frequency" in L202 with "hailstorm frequency".

**Reviewer Comment 2.11** — Figure 6: One of the authors' research questions is to investigate whether there are changes in the vertical wind shear surrounding hailstorms. The change in the vertical wind shear is not shown on Figures 6 and 7, only wind speed at 400 hPa in the current climate. It is shown in Figure 9 but I think it would already be useful to see it for the most intense storms.

**Reply 2.11**: In Fig. 9 the wind shear distribution is also indicated for the most intense storm instances (those yielding >30 mm). No changes were made to the manuscript regarding this comment.

**Reviewer Comment 2.12** — I'm wondering if the authors have considered kinematics in more detail. For example, depending on the hailstone trajectory it could be ejected earlier and not grow as large (e.g. Allen et al. 2020). The rain rate and wind gust panels are nice complementary information but are not specific to the dynamics/environment of hailstorms.

**Reply 2.12**: See Reply 2.3.

**Reviewer Comment 2.13** — L394-395. I don't agree with this conclusion in the way it is written. The level of the zero-degree isotherm does play a key role, otherwise we'd see large hail more often in the tropics. I think a more plausible hypothesis is that in the future scenario increased melting is offset by the increase in the typical maximum size hailstones can reach. I'd suggest rewording to not disregard the importance of melting.

**Reply 2.13**: See Reply 2.14.

**Reviewer Comment 2.14** — L396: I see now that in the subsequent lines the authors mention the hypothesis above and explain it in more detail. I'd suggest deleting lines L394-395 as it could be misleading.

**Reply 2.14**: Agreed, deleted as suggested.

**Reviewer Comment 2.15** — L434: Convective available potential energy?

**Reply 2.15**: Replaced "convective potential" with "CAPE".

**Reviewer Comment 2.16** — L444: Please remove unpublished studies. Furthermore, Wilhelm et al. (2024) showed an increase in hail days both north and south of the Alps whereas your results show a decrease in the number of hailstorms south of the Alps (Fig. 2c).

**Reply 2.16**: Concerning the references to unpublished studies, please refer to Reply 1.1a. The south Alpine domain in Wilhelm et al. (2024) encompasses areas (see Fig. 1 in their study) where our study finds both increases and decreases in future hailstorm frequencies. It must be noted here, that firstly, we show hailstorm frequencies and Wilhelm et al. (2024) show hail day frequencies, which are not directly comparable, and secondly Wilhelm et al. (2024) analyzes past trends, which might be different to the future trends in our simulations. Given the differing spatial domains, event definitions, and temporal scopes, we do not propose changes to the manuscript.

**Reviewer Comment 2.17** — L445: Where is it shown that hail environments change similarly in all subregions? The only geographic maps are Figs. 2 and 3. I could not see this in the supplement either, only inflow environments for storms in different regions. If the authors are referring to Figs. S6 – S16 I suggest they be specific that this is for the top 10% cases inflow environments not general trends per region.

**Reply 2.17**: Thank you for pointing out this missing reference, the sentence now reads: "An important result of this study is that the hail environments of the most intense hailstorms change similarly in all sub-regions — regardless of the local hailstorm frequency trend signal (see Fig. S6–S16)."

**Reviewer Comment 2.18** — As a general comment, I think the authors could be more precise what they are referring to when make their conclusions to avoid any confusion or misinterpretation of the results.

**Reply 2.18**: We acknowledge the importance of clarity and precision in the conclusions. The specific comments by both reviewers regarding the addition of references in the conclusion directly address this concern by prompting us to anchor key statements more explicitly. We consider these targeted revisions sufficient to resolve the broader issue raised in this general comment.

**Reviewer Comment 2.19** — L458–466: It's nice to see that the authors have openly discussed the limitations of this work. I think the fact that the changes in hail frequency/size/intensity etc will also be dependent on the actual level of warming is important to mention. In this study, the authors base all their results on 3 K of global warming which is a real possibility. Nevertheless, I think it should be mentioned that the results are valid assuming a 3 K warming and that the actual level of warming will depend on different factors (e.g., future emissions).

**Reply 2.19**: We've inserted the following paragraph on L467: "The results presented in this study are based on simulations assuming a global mean temperature increase of 3 K, representing a plausible high-end warming scenario (for more detail, see Thurnherr et al., 2025). It is important to note that projected changes in hail frequency, size, and intensity are contingent on the level of warming realized, which in turn depends on future greenhouse gas emissions and socio-economic developments (see e.g., Seneviratne et al., 2021). Thus, our findings should be interpreted within the context of this single 3 K warming scenario."

**References**

Adams-Selin, R., 2025: The Quasi-Stochastic Nature of Hail Growth: Hail Trajectory Clusters in Simulations of the Kingfisher, Oklahoma, Hailstorm. *Monthly Weather Review*, **153 (1)**, 67–87, DOI: `10.1175/MWR-D-23-0233.1`.

Adams-Selin, R. D. and C. L. Ziegler, 2016: Forecasting hail using a one-dimensional hail growth model within WRF. *Mon. Weather Rev.*, **144 (12)**, 4919–4939, DOI: `10.1175/MWR-D-16-0027.1`.

Adlerman, E. J. and K. K. Droegemeier, 2002: The sensitivity of numerically simulated cyclic meso-cyclogenesis to variations in model physical and computational parameters. *Monthly Weather Review*, **130 (11)**, 2671–2691, DOI: `10.1175/1520-0493(2002)130<2671:TSONSC>2.0.CO;2`.

Battaglioli, F., P. Groenemeijer, T. Púčik, M. Taszarek, U. Ulbrich, and H. Rust, 2023: Modeled multidecadal trends of lightning and (very) large hail in Europe and North America (1950–2021). *J. Appl. Meteorol. Clim.*, **62 (11)**, 1627–1653, DOI: `10.1175/JAMC-D-22-0195.1`.

Brennan, K. P., M. Sprenger, A. Walser, M. Arpagaus, and H. Wernli, 2024: An object-based and Lagrangian view on an intense hailstorm day in Switzerland as represented in COSMO-1E ensemble hindcast simulations. *EGUsphere [preprint]*, 1–31, DOI: `10.5194/egusphere-2024-2148`.

Brimelow, J. C., W. R. Burrows, and J. M. Hanesiak, 2017: The changing hail threat over North America in response to anthropogenic climate change. *Nat. Clim. Change*, **7 (7)**, 516–522, DOI: 10.1038/nclimate3321.

Cui, R., I. Thurnherr, P. Velasquez, K. Brennan, M. Leclair, A. Mazzoleni, T. Schmid, H. Wernli, and C. Schär, 2024: A european hail and lightning climatology from an 11-year kilometer-scale regional climate simulation. *J. Geophys. Res. Atmos. [preprint]*, DOI: 10.22541/essoar.173120326.69513564/v1.

Feldmann, M., M. Blanc, K. P. Brennan, I. Thurnherr, P. Velasquez, O. Martius, and C. Schär, 2025: European supercell thunderstorms – an underestimated current threat and an increasing future hazard. *preprint*, DOI: 10.48550/arXiv.2503.07466.

Gensini, V. A., W. S. Ashley, A. C. Michaelis, A. M. Haberlie, J. Goodin, and B. C. Wallace, 2024: Hailstone size dichotomy in a warming climate. *npj Clim. Atmos. Sci.*, **7 (1)**, 1–10, DOI: 10.1038/s41612-024-00728-9.

Mallinson, H., S. Lasher-Trapp, J. Trapp, M. Woods, and S. Orendorf, 2023: Hailfall in a possible future climate using a pseudo-global warming approach: Hail characteristics and mesoscale influences. *J. Climate*, **37 (2)**, 527–549, DOI: 10.1175/JCLI-D-23-0181.1.

Nowotarski, C. J., J. M. Peters, and J. P. Mulholland, 2020: Evaluating the effective inflow layer of simulated supercell updrafts. *Monthly Weather Review*, **148 (8)**, 3507–3532, DOI: 10.1175/MWR-D-20-0013.1.

Peters, J. M., H. Morrison, C. J. Nowotarski, J. P. Mulholland, and R. L. Thompson, 2020a: A formula for the maximum vertical velocity in supercell updrafts. *Journal of the Atmospheric Sciences*, **77 (11)**, 3747–3757, DOI: 10.1175/JAS-D-20-0103.1.

Peters, J. M., C. J. Nowotarski, and G. L. Mullendore, 2020b: Are supercells resistant to entrainment because of their rotation? *Journal of the Atmospheric Sciences*, **77 (4)**, 1475–1495, DOI: 10.1175/JAS-D-19-0316.1.

Prein, A. F., R. M. Rasmussen, D. Wang, and S. E. Giangrande, 2021: Sensitivity of organized convective storms to model grid spacing in current and future climates. *Philosophical Transactions of the Royal Society A: Mathematical, Physical and Engineering Sciences*, **379 (2195)**, 20190546, DOI: 10.1098/rsta.2019.0546.

Prein, A. F. and A. J. Heymsfield, 2020: Increased melting level height impacts surface precipitation phase and intensity. *Nat. Clim. Change*, **10 (8)**, 771–776, DOI: 10.1038/s41558-020-0825-x.

Pruppacher, H. R. and J. D. Klett, 2010: *Microphysics of clouds and precipitation*, 2nd ed., Atmospheric and Oceanographic Sciences Library, Springer Netherlands.

Raupach, T. H., O. Martius, J. T. Allen, M. Kunz, S. Lasher-Trapp, S. Mohr, K. L. Rasmussen, R. J. Trapp, and Q. Zhang, 2021: The effects of climate change on hailstorms. *Nat. Rev. Earth Environ.*, **2 (3)**, 213–226, DOI: 10.1038/s43017-020-00133-9.

Seneviratne, S. I., X. Zhang, M. Adnan, W. Badi, C. Dereczynski, A. D. Luca, S. Ghosh, I. Iskandar, J. Kossin, S. Lewis, F. Otto, P. Pinto, M. Satoh, S. M. Vicente-Serrano, M. Wehner, and B. Zhou, 2021: Weather and climate extreme events in a changing climate. In climate change 2021: The physical science basis. Contribution of working group I to the sixth assessment report of the intergovernmental panel on climate change. *Cambridge University Press, Cambridge, United Kingdom and New York, NY, USA*, 1513–1766, DOI: 10.1017/9781009157896.013.

Thurnherr, I., R. Cui, P. Velasquez, H. Wernli, and C. Schär, 2025: The effect of 3°C global warming on hail over Europe. *preprint*, DOI: 10.22541/au.173809555.59545480/v1.

Wilhelm, L., C. Schwierz, K. Schröer, M. Taszarek, and O. Martius, 2024: Reconstructing hail days in Switzerland with statistical models (1959–2022). *Nat. Hazards Earth Syst. Sci.*, **24 (11)**, 3869–3894, DOI: 10.5194/nhess-24-3869-2024.

---

## Author Response (AR2)

**Round two: Author's response for paper egusphere-2025-918**

**Insights from hailstorm track analysis in European climate change simulations**

by Killian P. Brennan, Iris Thurnherr, Michael Sprenger, and Heini Wernli

July 14, 2025
* * *
**Reviewer 1**

**Overview**

I thank the authors for their detailed responses. My main comments are resolved with the additional discussion included in the new manuscript. I only have a few optional suggestions remaining. Congratulations for this excellent article.

**Reply**: Thank you for your positive feedback and for acknowledging the revisions made. We are pleased to hear that the main comments have been satisfactorily addressed. Below, we respond to your remaining optional suggestions and indicate how we have incorporated them into the revised manuscript where applicable.

**Reviewer Comment 1.1** — I agree that you used the current state of the art, the comment was not meant as criticism about your approach. Just to expand on this aspect, I still believe you underestimate the limitations with respect to under-resolved updrafts. Adams-Selin (2025) and Fischer et al. (2025) recently emphasized how sensitive 3D hail trajectories are to updraft characteristics. I at least see it as a possibility that trends of certain hail sizes cannot be represented in your study because of the limit in realistic updrafts. Also, I'm not sure Prein et al. (2021) can be used as a strong support here. As you mention, they found that updraft characteristics continue to improve at resolution $< 4\,\text{km}$ and they did not look in detail at hail production or hailstorms in particular, which tend to be non-MSC. However, that's just my opinion, I'm ok with the manuscript additions above if you think it makes the limitations clear enough.

**Reply 1.1**: Thank you for the clarification. We agree that under-resolved updrafts remain a fundamental limitation in convection-permitting models and acknowledge the relevance of recent work by Adams-Selin (2025) and Fischer et al. (2025) in this context. Our intention was not to downplay this issue but to contextualize it within the current modeling capabilities. While we maintain that the manuscript sufficiently communicates the key limitations, we recognize the merit

of your perspective and appreciate the opportunity to reflect on this aspect more deeply. While Adams-Selin (2025) was already included in the discussion on limitations (L480) we've expanded the limitations with a reference of Fischer et al. (2025). Please also refer to Reply 1.3 in this document. We are of the opinion that the current state of the limitations section (L477–501) adequately addresses all points raised by the reviewers concerning model resolution, updraft realism, and associated implications for hail simulation.

**Additional comments**

**Reviewer Comment 1.2** — Line 6 and throughout: Is the italics for numbers intended?

**Reply 1.2**: Thank you for pointing this out. The use of italics for numbers was not intentional and has been corrected throughout the abstract.

**Reviewer Comment 1.3** — Lines 57-58: Would it make sense to put this shorter part before (iii)? At least to me it seems logical to end with the approach you are using. Just a suggestion.

**Reply 1.3**: Thank you for the suggestion. We agree that this ordering improves the logical flow and have swapped points (iii) and (iv) accordingly.

**Reviewer Comment 1.4** — Lines 303-311: As mentioned in line 123, hailcast does not take horizontal advection of hail into account so the landing position relative to the updraft might not be realistic. Fig. 5a also indicates this as hail mostly falls directly under the track. In other words, the relative horizontal transport of hail and rain (size sorting) is not fully represented. Yet here you seem to take a roughly correct position of hail relative to rain as granted. See radar studies or 3D hail trajectory simulations for more realistic fall locations of hail. I think you can still include this part given you look at relative differences. However, the caveat should I think be mentioned in the context of this paragraph.

**Reply 1.4**: Thank you for this thoughtful comment. As stated on line L275f: "Graupel is explicitly included in the COSMO microphysics and is subject to horizontal advection, which, however, results in only a slight offset of the graupel maximum from the storm center to the left relative to cell movement (Fig. 5a). The location of the graupel maximum provides an upper limit on potential hail advection, as graupel has a lower terminal velocity than the smallest hailstones, allowing more time for horizontal advection." The location of the graupel relative to the storm's center offers an upper limit to the expected horizontal advection of the hail within our simulations — in a storm-resolving simulation, it might be possible that hail (and graupel) is transported over larger distances. We believe no additional changes to the manuscript are necessary.

**Reviewer Comment 1.5** — Lines 497-501: I think it should also be mentioned that you only simulated a 10-year period, so changes in high-end hailstorms might not be represented, especially not at a regional level.

**Reply 1.5**: Thank you for the comment. We've added a sentence to the limitations to reflect this (L502): "In addition, the analysis is based on two 11-year periods, which may be insufficient to robustly capture changes in rare, high-end hail events, particularly at regional scales."

**Reviewer 2**

**Summary**

The authors have made some changes based on my comments, but some were not addressed. The authors sometimes claim this is due to lack of data availability, but they could be more creative in this respect. I also find myself going between this study and the Cui et al. and Thurnherr et al. preprints a lot. In the current state, I think that it may be difficult for other scientists to follow the paper outside of their field, which I would find a pity. Nevertheless, I leave it up to authors on whether they wish to make further revisions to improve the quality of their study and make it accessible to a wider audience.

**Reply**:  Thank you for your continued engagement and constructive feedback. We acknowledge that some of your earlier suggestions were not implemented, primarily due to data limitations, but we appreciate the encouragement to explore alternative approaches. We have reorganized Sect. 2.1 to make the methods more accessible.

**Comments**

**Reviewer Comment 2.1**  —  Regarding assessing changes in CI frequency in current/future climate, it is said that no radar reflectivity data is available, however on L116 it is said that precipitation fields are saved every 5 minutes. Furthermore, Cui et al. (2024) build a lightning climatology in COSMO using the lightning potential index (LPI). Therefore, there are indeed data available to use as a proxy for CI. This would be a very basic analysis that will enable further interpretation of the results. In a response to a comment, the reviewers mention that a revised version of Thurnherr et al. (2025) (not available publicly) found changes in CIN in the region with decreasing hailstorm frequency. This should be mentioned in the manuscript.

**Reply 2.1**:  Thank you for this constructive comment. While precipitation and lightning fields are available in the model output, we don't expect that they provide more insight on the question of convection initiation without applying a similar tracking algorithm as applied to the hail fields in this study. Without such a tracking, it is not possible to clearly differentiate between the effect of changing frequencies (i.e. changing convection initiation) or changing amounts of precipitation/lightning per convective storm. It is the strength of the tracking algorithm applied in this study that we were able to differentiate between such effects. And it is beyond the scope of this study to develop and verify a tracking of lightning and precipitation cells, but future studies could explore this utilizing our published hail tracking code. The following sentence was added (L209): "In regions where hailstorm frequency decreases — such as central France — CAPE is reduced and CIN increases in the seasonal Eulerian mean, consistent with less frequent convective initiation (Thurnherr et al., 2025)."

**Reviewer Comment 2.2**  —  The authors' response to comment 2.16 confuses me. They now claim that their study and Wilhelm et al. (2024) are not comparable. Why are they then using the

study to support their conclusions? Please either remove this citation as support for their results or discuss further the nuances.

**Reply 2.2**: Thank you for raising this point. We agree that the comparison with Wilhelm et al. (2024) should be more clearly qualified. The section on L463f now reads: "The findings in this study align with past trend analyses in the northern Alpine domain (1959–2022, Wilhelm et al., 2024), which reported similar regional signals. South of the Alps, Wilhelm et al. (2024) also identified a positive past trend in yearly haildays. In our projections, both positive and negative changes in hailstorm frequency occur in close spatial proximity within the southern Alpine domain, with increases along the southern fringes of the main Alpine crest and a pronounced localized decrease in the Po valley (Fig. 2c). Observational studies such as Manzato et al. (2022) support this signal, showing a slight negative past trend in northeastern Italy, which is consistent with our projected localized decrease in hailstorm frequency in that area. These findings are further consistent with projections from studies such as Raupach et al. (2021), Thurnherr et al. (2025), and Feldmann et al. (2025), which report comparable regional trends throughout the remaining simulation domain."

**Reviewer Comment 2.3** — L72: This will likely get changed by the typesetters, but it would be better to write "The simulations in Thurnherr et al. (2025)"

**Reply 2.3**: Thank you, we've changed this as you've suggested.

**Reviewer Comment 2.4** — L138: "Be" is repeated.

**Reply 2.4**: Thank you for pointing this out. This has been rectified.

**References**

Adams-Selin, R., 2025: The Quasi-Stochastic Nature of Hail Growth: Hail Trajectory Clusters in Simulations of the Kingfisher, Oklahoma, Hailstorm. *Mon. Weather Rev.*, **153 (1)**, 67–87, DOI: `10.1175/MWR-D-23-0233.1`.

Feldmann, M., M. Blanc, K. P. Brennan, I. Thurnherr, P. Velasquez, O. Martius, and C. Schär, 2025: European supercell thunderstorms – an underestimated current threat and an increasing future hazard. *preprint*, DOI: `10.48550/arXiv.2503.07466`.

Fischer, J., M. Kunz, K. Lombardo, and M. R. Kumjian, 2025: Hail Trajectories in a Wide Spectrum of Supercell-Like Updrafts. *J. Atmos. Sci.*, **82 (7)**, 1403–1422, DOI: `10.1175/JAS-D-24-0222.1`.

Manzato, A., A. Cicogna, M. Centore, P. Battistutta, and M. Trevisan, 2022: Hailstone characteristics in northeast Italy from 29 years of hailpad data. *J. Appl. Meteorol. Clim.*, **61 (11)**, 1779–1795, DOI: `10.1175/JAMC-D-21-0251.1`.

Raupach, T. H., O. Martius, J. T. Allen, M. Kunz, S. Lasher-Trapp, S. Mohr, K. L. Rasmussen, R. J. Trapp, and Q. Zhang, 2021: The effects of climate change on hailstorms. *Nat. Rev. Earth Environ.*, **2 (3)**, 213–226, DOI: `10.1038/s43017-020-00133-9`.

Thurnherr, I., R. Cui, P. Velasquez, H. Wernli, and C. Schär, 2025: The effect of 3°C global warming on hail over Europe. *preprint*, DOI: `10.22541/au.173809555.59545480/v1`.

Wilhelm, L., C. Schwierz, K. Schröer, M. Taszarek, and O. Martius, 2024: Reconstructing hail days in Switzerland with statistical models (1959–2022). *Nat. Hazards Earth Syst. Sci.*, **24 (11)**, 3869–3894, DOI: 10.5194/nhess-24-3869-2024.